# Sonic hedgehog-heat shock protein 90β axis promotes the development of nonalcoholic steatohepatitis in mice

Weitao Zhang[1,2], Junfeng Lu[3], Lianshun Feng[2], Hanyue Xue[2], Shiyang Shen[2], Shuiqing Lai[4], PingPing Li [5], Ping Li[2], Jian Kuang[4], Zhiwei Yang [6] ✉ & Xiaojun Xu [1,2] ✉

Sonic hedgehog (SHH) and heat shock protein 90β (HSP90β) have been implicated in nonalcoholic steatohepatitis (NASH) but their molecular mechanisms of action remain elusive. We find that HSP90β is a key SHH downstream molecule for promoting NASH process. In hepatocytes, SHH reduces HSP90β ubiquitylation through deubiquitylase USP31, thus preventing HSP90β degradation and promoting hepatic lipid synthesis. HSP90β significantly increases in NASH mouse model, leading to secretion of exosomes enriched with miR-28-5p. miR-28-5p directly targets and decreases Rap1b levels, which in turn promotes NF-κB transcriptional activity in macrophages and stimulates the expression of inflammatory factors. Genetic deletion, pharmacological inhibition of the SHH-HSP90β axis, or delivery of miR-28-5p to macrophages in the male mice liver, impairs NASH symptomatic development. Importantly, there is a markedly higher abundance of miR-28-5p in NASH patient sera. Taken together, the SHH-HSP90β-miR-28-5p axis offers promising therapeutic targets against NASH, and serum miR-28-5p may serve as a NASH diagnostic biomarker.

In the past 40 years, nonalcoholic fatty liver disease (NAFLD) has emerged as the most common chronic liver disease (with a global prevalence in adults at about 25%). NAFLD includes nonalcoholic fatty liver (NAFL) and nonalcoholic steatohepatitis (NASH)[1]. Only 4% of patients with NAFL develop cirrhosis or liver cancer, while patients with NASH are at a greater risk of developing cirrhosis, with a rate of more than 20%[2,3]. NASH is the progressive form of NAFLD and is characterized by liver steatosis, inflammation, hepatocellular injury, and varying degrees of fibrosis[4]. Hepatic steatosis, the excessive

accumulation of fat in hepatocytes, occurring in NAFL and NASH[5], renders the liver cells more susceptible to injury, inflammation, and apoptosis[6]. Triglyceride (TG) is the main type of fat stored in the fatty liver without exerting hepatotoxicity but some lipid types accumulated in the fatty liver, including free fatty acids (FFA), diacylglycerol, oxysterols, cholesterol, and phospholipids can damage hepatocytes[7]. Fatty acids in the steatotic liver are important signaling molecules that link inflammation through endoplasmic reticulum stress[8]. Inflammation is the body's physiological response stimulated upon tissue injury

[1]Department of Pharmacy, The Fourth Affiliated Hospital, Zhejiang University School of Medicine, Yiwu, Zhejiang, China; Center for Innovative Traditional Chinese Medicine Target and New Drug Research, International Institutes of Medicine, Zhejiang University, Yiwu, Zhejiang, China. [2]State Key Laboratory of Natural Medicines, China Pharmaceutical University, 210009 Nanjing, Jiangsu, China. [3]First Department of Liver Disease, Beijing You'An Hospital, Capital Medical University, Beijing 100069, China. [4]Guangdong Provincial People's Hospital, Guangdong Academy of Medical Sciences, Guangzhou, Guangdong 510080, China. [5]State Key Laboratory of Bioactive Substance and Function of Natural Medicines, Institute of Materia Medica, Chinese Academy of Medical Sciences & Peking Union Medical College, Diabetes Research Center of Chinese Academy of Medical Sciences, Beijing 100050, China. [6]Institute of Laboratory Animal Science, Chinese Academy of Medical Sciences (CAMS) & Peking Union Medical Collage (PUMC), Beijing 100021, PR China. ✉e-mail: yangzhiwei@cnilas.pumc.edu.cn; xiaojunxu@zju.edu.cn

or infection. It is a key process in NASH. It coordinates cellular defense mechanisms and tissue repair by secreting various inflammatory mediators, like cytokines and chemokines. However, over time, persistent chronic inflammation exacerbates tissue injury causing an abnormal wound-healing response. The hepatic inflammatory response is an important driving force for the transformation from NAFLD to NASH, as it promotes continuous liver fibrosis, eventually leading to liver cirrhosis[4].

Several studies suggest that heat shock protein 90β (HSP90β, encoded by *HSP90AB1*) is closely related to liver steatosis. For example, the expression of HSP90β is high in the liver of NAFLD patients and obese mice and can promote the de novo lipid synthesis in the liver by stabilizing SREBPs[9,10]. Serum HSP90β levels in children with overweight or obesity are significantly elevated, while there is no difference in HSP90α expression (encoded by *HSP90AA1*)[11]. Currently, a tissue-specific *Hsp90β* knockout mouse model is lacking for assessing the subtype-specific function. Moreover, the regulatory mechanism underlying pathological HSP90β overexpression in the disease state is elusive.

In addition to its effect on liver lipid metabolism, HSP90 affects intercellular communication through exosomal secretion. HSP90 directly interacts with the membrane through the evolutionarily conserved amphiphilic helix structure, thus regulating exosomal release by promoting the fusion of multivesicular bodies (MVB) with the plasma membrane[12]. Exosomes are single-membrane vesicles released into the extracellular matrix after the fusion of intracellular multivesicles and the cell membrane. Almost all cell types can produce and release exosomes. It is a nanoscale lipid inclusion structure with a diameter ranging from 30 to 100 nm, internally enclosing proteins, mRNA, microRNAs (miRNAs), and other substances. Exosomes are involved in material transport and information exchange between hepatocytes, other cell types in the liver, and even other tissues and organs. Exosome-mediated signaling leads to several liver biological responses, including inflammation, angiogenesis, proliferation, and tissue remodeling. Therefore, exosomes play an important role in several liver diseases[13]. Whether pathological overexpression of HSP90β affects liver function and promotes the process of NASH by affecting the type and abundance of miRNAs in liver secretion necessitates clarification.

The hedgehog (HH) pathway is a conserved and highly complex signaling transmission involved in cell growth, survival, and fate determination. Three ligands are involved in the HH pathway, namely sonic hedgehog (SHH), Indian hedgehog (IHH), and desert hedgehog (DHH). Their tissue localization and molecular functions are different. The HH pathway can be simplified into the four following basic components: HH ligand, receptor patched (PTCH), signal sensor smoothened (SMO), and effector transcription factor GLI. GLI protein belongs to the Kruppel-like family of transcription factors and contains a zinc finger DNA binding domain[14]. There are three Gli proteins in mammals–GLI1, GLI2, and GLI3. In the absence of the HH ligand, SMO is inactivated and the transcription factor, GLI, is prevented from entering the nucleus by binding to an inhibitory complex composed of the fused kinase (FU)[15], an inhibitor of fused (SUFU), and Costal-2 (mammalian ortholog KIF7)[16]. When SMO is activated, GLI is separated from SUFU[17], it enters the nucleus and promotes the transcription of downstream genes. In addition to the above classical HH signaling pathways, there are two types of noncanonical HH signaling pathways. The noncanonical HH signaling pathway includes PTCH1 triggering apoptosis by activating caspase-3[18] and SMO promoting glycolysis and metabolism in the muscle and adipose tissues through the calcium-AMPK kinase axis[19].

Healthy hepatocytes negligibly express SHH but under stressed conditions, including endoplasmic reticulum stress and lipotoxicity caused by free fatty acid accumulation, these rapidly up-regulate the expression of SHH[20,21]. When liver damage is prevented (such as by caspase-2 deletion), hepatocytes can be protected from free fatty acid-induced apoptosis and SHH expression is not induced[22]. Therefore, SHH expression is closely related to the damage of liver cells. In clinical practice, the activation of the SHH pathway is associated with the risk of liver cell injury/death, liver inflammation, the severity of liver fibrosis, and the incidence rate of liver-related diseases along with the worsening of mortality rates[23–25]. Several animal models of NAFLD have been used to confirm the activation of the SHH pathway, which correlates positively with the severity of steatohepatitis and liver fibrosis[26,27]. Consistently, the activation of SHH correlates negatively with the prognosis of NASH patients[28]. Interestingly, *Leptin* deficient *ob/ob* mice are obese but the possibility of steatohepatitis and liver fibrosis decreases. One possible reason is that leptin activates SHH expression and a decrease in SHH activity in leptin-deficient *ob/ob* mice might lead to the inhibition of inflammation and fibrosis[29,30]. SHH regulates the expression of the long-chain noncoding RNA, *Hilnc* (HH-induced long noncoding RNA), thus participating in the regulation of liver lipid metabolism induced by high fat[31]. SHH promotes the infiltration of inflammatory cells in the liver by inducing the expression of osteopontin[31], and fibrosis by changing the glucose metabolism of hepatic stellate cells (HSCs)[32,33]. However, the downstream key factors and the detailed molecular mechanism of SHH signaling action in the process of NASH remain unclear.

In this study, we demonstrated that an abnormally high expression of SHH was an important regulatory factor affecting the pathological increase in HSP90β. During NASH development, HSP90β promoted lipid synthesis in hepatocytes by activating the SREBPs pathway. HSP90β changed the variety and abundance of miRNAs in liver exosomes, in turn promoting the inflammatory responses of liver macrophages, secretion of extracellular matrix (ECM), and fibrosis.

## Results

### SHH inhibitor protects mice from HFFC-diet-induced fatty liver, insulin resistance, inflammation, and fibrosis

In the liver section of NASH patients, SHH was highly expressed compared to that of the healthy donors (Fig. 1a). We tested whether pharmacological inhibition of SHH could alleviate NASH symptoms in mouse models. Cyclopamine (CP), a Smoothened (SMO) inhibitor[34,35] was used to block SHH signaling transduction (Fig. 1b). Under continuous high fat, high fructose, and high cholesterol diet (HFFC) diet feeding, Shh concentration increased in the serum. In contrast, CP did not affect serum Shh concentration (Supplementary Fig. 1a) but drastically reduced *Gli1* expression (Supplementary Fig. 1b), indicating that CP was effective in the animal model. HFFC-fed mice absorbed lipids and deposited them in various organs including adipose tissues, liver, etc. Compared to vehicle-fed control mice, the administration of CP significantly reduced liver body ratio, the percentage of visceral adipose tissue (sWAT), epididymal white adipose tissue (eWAT), and increased brown adipose tissue (BAT) (Supplementary Fig. 1c–f). The effect of fat reduction was directly reflected in the weight loss of mice in the CP group (Fig. 1c). Fat was stored in cells in the form of lipid droplets. HFFC diet significantly increased the content of lipid droplets in the liver, while CP reversed this phenomenon, i.e., causing a reduction in the number of lipid droplets in the liver of obese mice (Fig. 1d).

The homeostasis of lipid metabolism is largely maintained by regulating exogenous absorption, endogenous synthesis, lipid degradation, and transformation. When the body's adipose tissues store excessive lipids, lipids such as cholesterol and TG are continuously released into the blood, resulting in lipid disorders, atherosclerosis, and other complications[36]. After prolonged feeding on the HFFC diet, total cholesterol (TC)[35], TG, and low-density lipoprotein cholesterol (LDL-c) levels in the serum increased significantly. CP significantly mitigated these metabolic abnormalities (Supplementary Fig. 1g–i). CP did not affect the slightly increased HDL-c levels in HFFC-fed mice (Supplementary Fig. 1j).

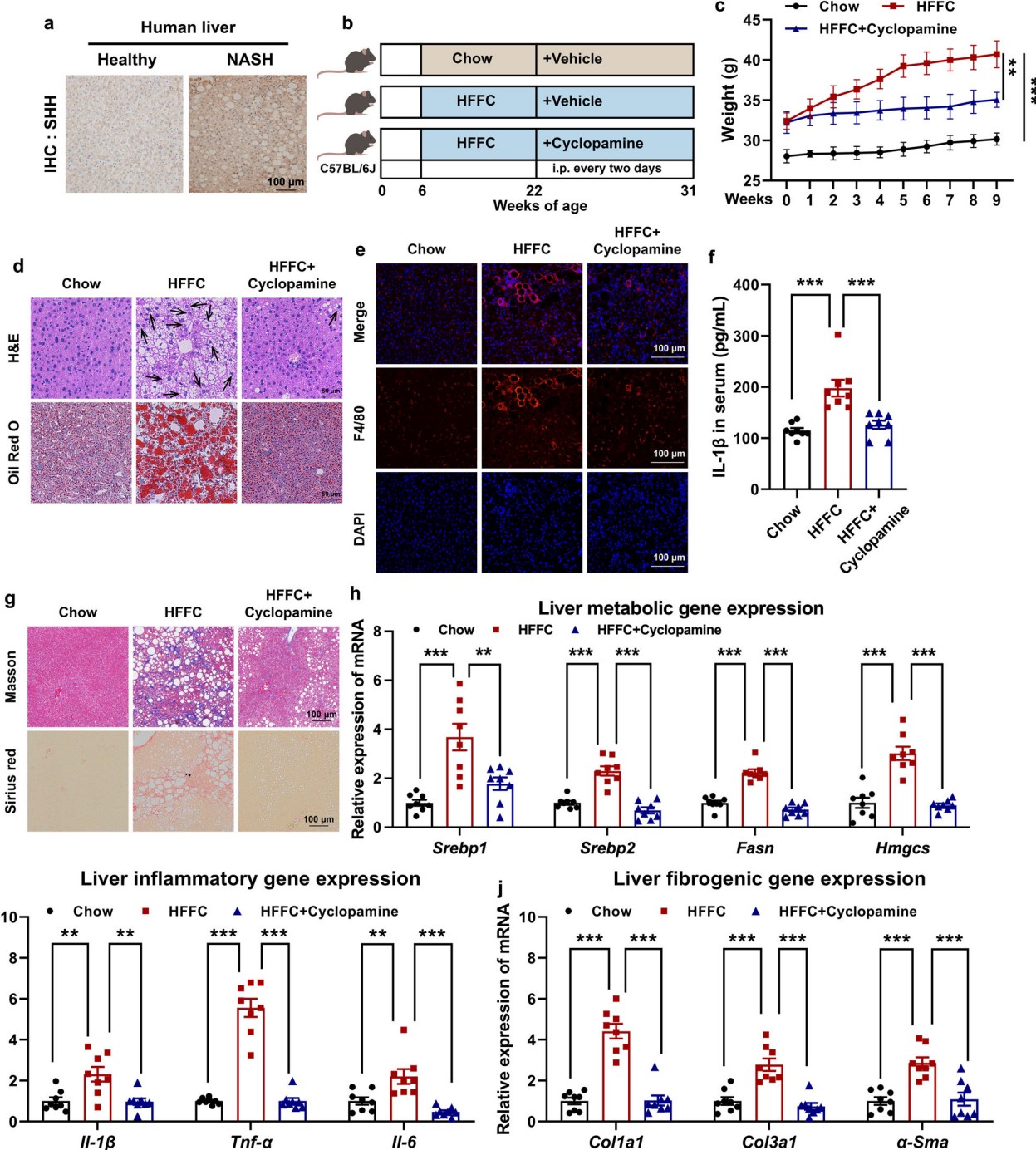

**Fig. 1 | Cyclopamin protects mice from HFFC diet-induced fatty liver, inflammation and fibrosis. a** Representative immunohistochemistry staining of SHH in the livers of healthy donors and NAFLD patients. **b** Experimental scheme of evaluating the therapeutic effects of cyclopamine in HFFC-fed mice. **c** Body weight of chow-fed mice or HFFC-fed mice administrated with vehicle or cyclopamine ($n = 8$ mice per group; body weight of 9th week, chow versus HFFC, $P = 6.0\text{E-}06$, HFFC versus HFFC+cyclopamine, $P = 0.0055$). **d** Formalin fixed paraffin embedded (FFPE) liver sections were stained with hematoxylin eosin (H&E) and frozen liver sections were stained with oil red O (ORO), respectively. Each arrow indicated hepatocyte ballooning. **e** Representative F4/80 stained liver sections. **f** Serum IL-1β levels of chow or HFFC-fed mice ($n = 8$ mice per group; chow versus HFFC, $P = 4.9\text{E-}05$, HFFC versus HFFC+cyclopamine, $P = 2.7\text{E-}04$). **g** Representative Masson and Sirius red stained liver sections. Hepatic expression of (**h**) metabolic (for *Srebp1*, chow versus HFFC, $P = 5.4\text{E-}05$, HFFC versus HFFC+cyclopamine, $P = 0.0021$; for *Srebp2*, chow versus HFFC, $P = 3.1\text{E-}06$, HFFC versus HFFC+cyclopamine, $P = 1.3\text{E-}07$; for *Fasn*, chow versus HFFC, $P = 7.2\text{E-}08$, HFFC versus HFFC+cyclopamine, $P = 2.1\text{E-}08$; for *Hmgcs*, chow versus HFFC, $P = 1.2\text{E-}06$, HFFC versus HFFC+cyclopamine, $P = 5.7\text{E-}07$), (**i**) inflammatory (for *Il-1β*, chow versus HFFC, $P = 0.0022$, HFFC versus HFFC +cyclopamine, $P = 0.0018$; for *Tnf-α*, chow versus HFFC, $P = 2.0\text{E-}08$, HFFC versus HFFC+cyclopamine, $P = 2.0\text{E-}08$; for *Il-6*, chow versus HFFC, $P = 0.0033$, HFFC versus HFFC+cyclopamine, $P = 8.1\text{E-}05$), and (**j**) fibrogenic (for *Col1a1*, chow versus HFFC, $P = 4.8\text{E-}08$, HFFC versus HFFC+cyclopamine, $P = 5.2\text{E-}08$; for *Col3a1*, chow versus HFFC, $P = 3.6\text{E-}05$, HFFC versus HFFC+cyclopamine, $P = 5.7\text{E-}06$; for *α-Sma*, chow versus HFFC, $P = 1.1\text{E-}04$, HFFC versus HFFC+cyclopamine, $P = 2.0\text{E-}04$) genes were analyzed by qRT-PCR ($n = 8$ mice per group). Data are presented as mean ± SEM. *$P < 0.05$, **$P < 0.01$, ***$P < 0.001$, NS no significant difference, one-way ANOVA. Source data are provided in the Source Data file.

We further investigated whether the SHH inhibitor dampened other pathological outcomes of HFFC diet-fed mice. Compared with chow diet-fed mice, 25-weeks HFFC diet-induced hepatic inflammation, as evidenced by infiltration of F4/80 positive macrophages (Fig. 1e), serum Il-1β increase (Fig. 1f), and collagen fibrils formation representing mature fibrosis highlighted by Masson and Sirius red staining (Fig. 1g) were observed. CP protected mice from HFFC diet-induced pathological changes (Fig. 1d–g). Consistently, genes involved in hepatic lipogenesis, inflammation, and fibrosis decreased in CP-treated mice (Fig. 1h–j). Overall, these data suggested that inhibiting SHH in mice largely alleviated HFFC diet-induced hepatic steatohepatitis.

## Shh stimulates Hsp90β protein expression in the NASH model

HSP90 has long been regarded as a housekeeping gene and serves as a loading control in several experiments. However, in our previous work, we found that HSP90β promoted de novo lipogenesis[9,37]. These studies indicated that HSP90β may exhibit paralog-specific roles in pathological conditions. We then analyzed liver Hsp90 content in the vehicle or CP-treated HFFC-fed mice. CP did not affect the levels of the commonly used housekeeping genes including Gapdh, β-Actin, Tubulin, Hsp70, and Hsp90α but significantly decreased the levels of Hsp90β and the Shh downstream transcription factor, Gli1 (Fig. 2a). In comparison, HFFC-fed mice exhibited higher expression of Hsp90β and Gli1 (Fig. 2b). These results indicated that Shh may regulate Hsp90β expression. We further checked Hsp90β expression in isolated primary hepatocytes (HPs), Kupffer cells (KCs) and hepatic stellate cells (HSCs) from mice fed with chow or HFFC diet for 16 weeks. It turned out that Hsp90β only increased in primary hepatocytes (Fig. 2c). Likewise, Shh only promoted Hsp90β protein levels in HPs (Fig. 2d). In hepatocytes, Shh increased the levels of Hsp90β in a time- and dose-dependent manner but not that of the Hsp90α protein (Fig. 2e, f). In murine and human liver cell lines and primary hepatocytes, Shh-induced Hsp90β increase was blocked by CP (Fig. 2g–i). SMO knock-down also reversed the regulatory effect of Shh on Hsp90β (Fig. 2j, k). Hsp90β correlated positively with Gli1 expression in the presence of an Shh activator (Smoothened Agonist, SAG)[38] (Fig. 2l). Among healthy controls, NAFLD and NASH patients, the expression levels of HSP90β, but not HSP90α, was positively correlated with disease progression (Fig. 2m–o).

## Hsp90β functions downstream of Shh during NASH development

Next, we sought to address whether Hsp90β was a key downstream target driving hepatic steatohepatitis. We first generated hepatocellular Hsp90β knockout mice (Hsp90β^ΔHep) by crossing Hsp90β^fl/fl and Alb-Cre (Supplementary Fig. 2a) mice. Hsp90β levels in livers and various cell types were examined after genotyping (Supplementary Fig. 2b, c). The genotype of 18 mice for the subsequent experiments was confirmed (Supplementary Fig. 2d). To examine the effects of hepatocyte Hsp90β ablation on steatohepatitis, Hsp90β^fl/fl or Hsp90β^ΔHep mice were fed chow or HFFC diet starting at 8 weeks of age for another 33 weeks (Supplementary Fig. 3a). In HFFC-fed mice, serum Shh concentration increased significantly, while hepatocellular deletion of Hsp90β did not affect serum Shh concentration (Supplementary Fig. 3b). In HFFC-fed Hsp90β^ΔHep mice, liver TC/TG contents (Supplementary Fig. 3c, d) and serum TC, TG, LDL-c, Il-1β, ALT, and AST (Supplementary Fig. 3e–k) levels were all alleviated compared with the HFFC-fed Hsp90β^fl/fl mice. Moreover, NASH histological features including hepatic lipid accumulation (Supplementary Fig. 4a), infiltration of F4/80 positive macrophages (Supplementary Fig. 4b) and staining for fibrosis by Masson and Sirius red (Supplementary Fig. 4c) were markedly diminished in Hsp90β^ΔHep mice. Genes involved in inflammation (Supplementary Fig. 4d) and fibrosis (Supplementary Fig. 4e) also decreased in Hsp90β^ΔHep mice. Consistently, 17AAG, a

widely used Hsp90 inhibitor, exhibited similar efficacy as the Hsp90β knockout (Supplementary Fig. 5).

Next, we investigated the role of Hsp90β during NASH development following the overexpression of Shh in the liver. Hsp90β^fl/fl or Hsp90β^ΔHep mice were fed an HFFC diet starting at 8 weeks of age. Mice were injected with ADV-NC or ADV-Shh every 4 weeks via their tail vein (Fig. 3a). GFP signals were observed in the livers of all four groups of mice, indicating successful viral delivery (Supplementary Fig. 6a). When injected with ADV-Shh, both Hsp90β^fl/fl and Hsp90β^ΔHep mice exhibited increased liver and serum Shh levels and upregulated hepatic Gli1 expression (Supplementary Fig. 6c), indicating the successful expression of the carried genes (Fig. 3b, and Supplementary Fig. 6b). In Hsp90β^fl/fl mice, ADV-Shh accelerated hepatic steatohepatitis, including lipid accumulation in the liver (Fig. 3c), hepatic damage (Supplementary Fig. 6f, g), inflammation (Fig. 3d–f), and fibrosis (Fig. 3g, h). Hepatocyte-specific Hsp90β ablation almost completely reversed these trends (Fig. 3b–h). Taken together, these results demonstrated that hepatocellular ablation of Hsp90β prevented HFFC-induced hepatic steatosis, inflammation, and fibrosis, suggesting that Hsp90β was required for Shh-triggered NASH development.

## Shh promotes Hsp90β deubiquitylation through Usp31

A series of deleterious factors (including free fatty acids, inflammatory factors, high glucose, or insulin) that occur in NASH did not increase Hsp90β levels, implying that Shh was the main mediator of Hsp90β accumulation in hepatocytes in mouse NASH models (Supplementary Fig. 7a, b). To understand the mechanism underlying the upregulation of Hsp90β by Shh, we first analyzed the mRNA levels of Hsp90. Both Hsp90aa1 (encoding Hsp90α) and Hsp90ab1 (encoding Hsp90β) levels remained unchanged in the presence of the Shh inhibitor and activator (Supplementary Fig. 7c, d). Next, we performed a transcriptome analysis and found that after Shh treatment, pathways associated with ribosome, oxidative phosphorylation, neurodegenerative diseases, thermogenesis, NAFLD, and proteasome were significantly enriched (Supplementary Fig. 8a, b). Protein stability assay for Hsp90β suggested that Shh decreased the Hsp90β degradation rate in cycloheximide (CHX) treated cells (Fig. 4a). Hsp90β ubiquitylation was reduced in Hepa 1–6 cells treated with Shh and primary hepatocytes of HFFC diet-fed mice (Fig. 4b, c). In contrast, Smo siRNA or CP reversed Hsp90β ubiquitylation (Fig. 4d, and Supplementary Fig. 7e). These results suggested that the induction of Hsp90β protein by Shh was achieved through inhibition of ubiquitylation-proteasomal degradation. Decreased Hsp90β ubiquitylation could be due to either decreased E3 activity or enhanced deubiquitylase (DUB) activity. We subsequently performed an in vitro deubiquitylation assay to distinguish between these possibilities (Fig. 4e, and Supplementary Fig. 7f). Hsp90β ubiquitylation remained unchanged in cell lysates of primary hepatocytes from chow-diet mice or Hepa 1–6 cells and primary hepatocytes treated with control vehicle (Fig. 4e, and Supplementary Fig. 7f). However, it decreased substantially in cell lysates of primary hepatocytes from HFFC-diet mice or Hepa 1–6 cells and primary hepatocytes treated with Shh[39] (Fig. 4e, and Supplementary Fig. 7f). Taken together, these results suggested that Shh-treated cell lysates exhibited a higher DUB activity, thus protecting Hsp90β from ubiquitin-mediated proteasomal degradation. RNA-seq data showed that Shh treatment significantly increased 14 DUBs (Fig. 4f). Only one of them (Usp31) increased in the liver of the HFFC-fed NASH mouse model (Fig. 4g). As HFFC feeding induced Shh signaling in the liver (Fig. 2b, and Supplementary Fig. 1a, b), we reasonably speculated that Usp31 was the in vivo target of Shh signaling. In hepatocytes, Shh increased Usp31 gene expression in a time- and dose-dependent manner, which was suppressed by Shh inhibition (Fig. 4h, and Supplementary Fig. 8c). In Usp31 knocked-down cells (Supplementary Fig. 8d), Shh-induced Hsp90β deubiquitylation and protein enrichment were fully abrogated (Fig. 4i, j, and Supplementary Fig. 8e, f).

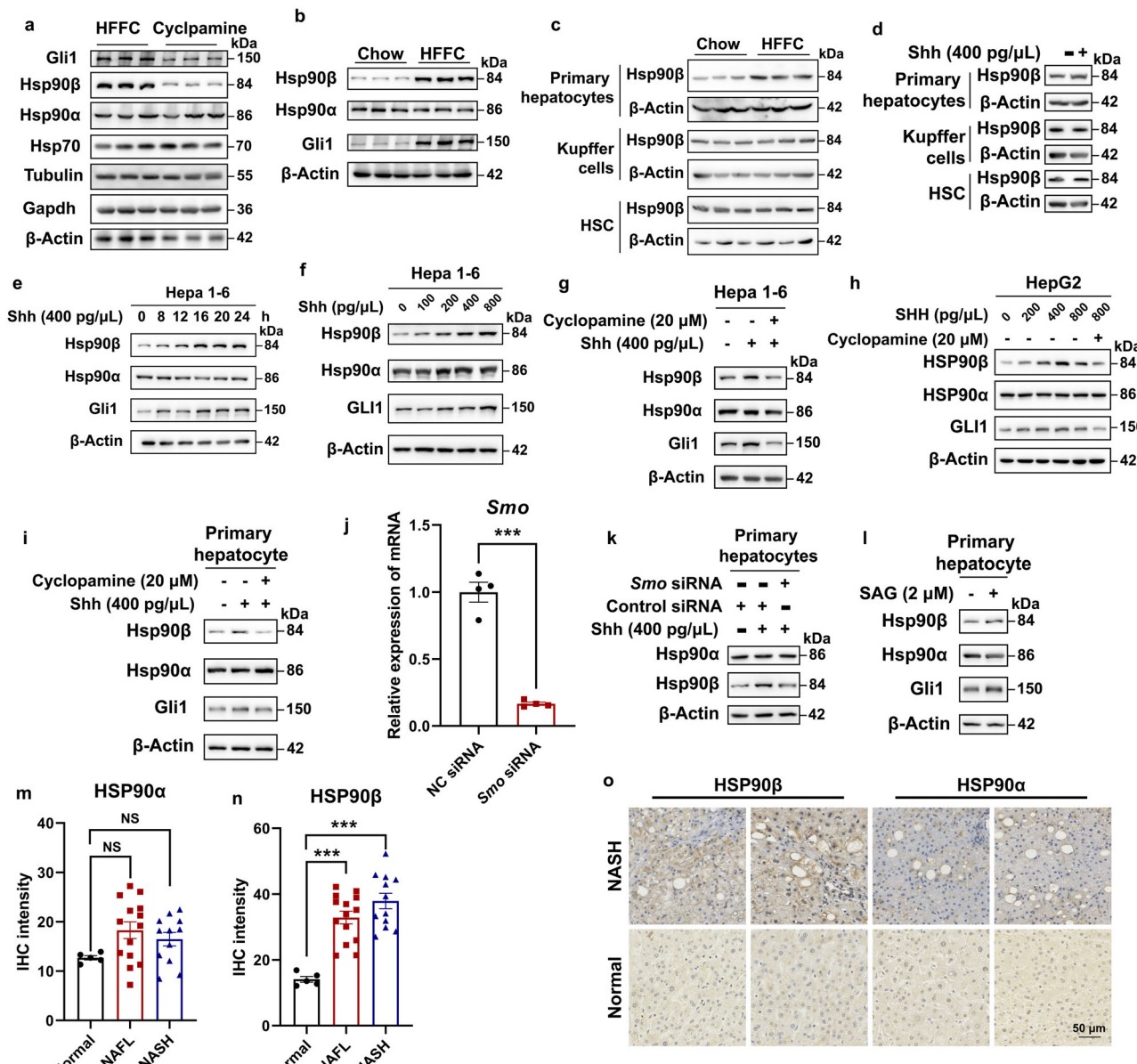

**Fig. 2 | Shh promotes Hsp90β expression in hepatocytes. a** Western blot analysis of hepatic protein samples from HFFC-diet mice treated with vehicle or cp. **b** Western blot analysis of hepatic protein samples from mice fed with a normal chow or HFFC diet. **c** Western blot analysis of primary hepatocyte, Kupffer cells and HSCs from mice fed with chow or HFFC diet. **d** Western blot analysis of primary hepatocyte treated with vehicle or Shh. Western blot analysis of Hepa 1–6 cells treated with (**e**) 400 pg/μL Shh for 1–24 h, (**f**) 0–800 pg/μL Shh for 24 h, or (**g**) 400 pg/μL Shh with or without 20 μM cyclopamine for 24 h. **h** Western blot analysis of HepG2 cells treated with 0–800 pg/μL Shh with or without 20 μM cyclopamine for 24 h. **i** Western blot analysis of primary hepatocytes treated with 400 pg/μL Shh with or without 20 μM cyclopamine for 24 h. **j** The knockdown efficiency of RNAi

targeting *Smo* assessed by qRT-PCR (*n* = 4 independent experiments per group; *NC* siRNA versus *Smo* siRNA, *P* = 3.1E-05). Western blot analysis of primary hepatocytes treated with (**k**) 400 pg/μL Shh after Smo knocked down or (**l**) 2 μM SAG. The IHC signal intensity of (**m**) HSP90α (normal versus NAFL, *P* = 0.083, normal versus NASH, *P* = 0.29) or (**n**) HSP90β (normal versus NAFL, *P* = 4.4E-04, normal versus NASH, *P* = 1.6E-05) in livers of healthy donors, NAFL and NASH patients. *n* = 5 participants in normal group. *n* = 14 participants in NAFL group. *n* = 12 participants in NASH group. **o** Representative immunohistochemistry staining of HSP90α or HSP90β in the livers of healthy donors and NASH patients. Data are presented as mean ± SEM. *P < 0.05, **P < 0.01, ***P < 0.001, NS no significant difference, one-way ANOVA. Source data are provided in the Source Data file.

Taken together, we concluded that Usp31 was a bona fide DUB of Hsp90β.

To identify the signal mediating Shh-induced *Usp31* over-expression, we first analyzed the promoter region of *Usp31*. The putative transcriptional factors were predicted by JASPAR[40] (Fig. 5a). We then transiently knocked down each of these transcriptional factors using siRNAs (Supplementary Fig. 9a) and observed that *Bmal1* reduced the expression of *Usp31* (Fig. 5b). We also observed BMAL1 upregulation in the liver sections of NASH patients (Fig. 5c). We then validated the binding of Bmal1 to the *Usp31* promoter

region. Electrophoretic mobility shift assay (EMSA) demonstrated that the 1727/1736 region (AGTCACGTGA) was the most likely Bmal1 binding motif within the fragment of the *Usp31* promoter (Fig. 5d, and Supplementary Fig. 9b, c). We further performed chromatin immunoprecipitation (ChIP) assays and found the recruitment of Bmal1 to the 1637−1796 bp region containing a predicted Bmal1 binding motif (Fig. 5e, and Supplementary Fig. 9b). In comparison, Shh did not increase the binding of Srebp1 to the *Usp31* promoter region containing the Srebp1 binding motif (Fig. 5e, and Supplementary Fig. 9b, d). Further, Shh increased Bmal1's nuclear

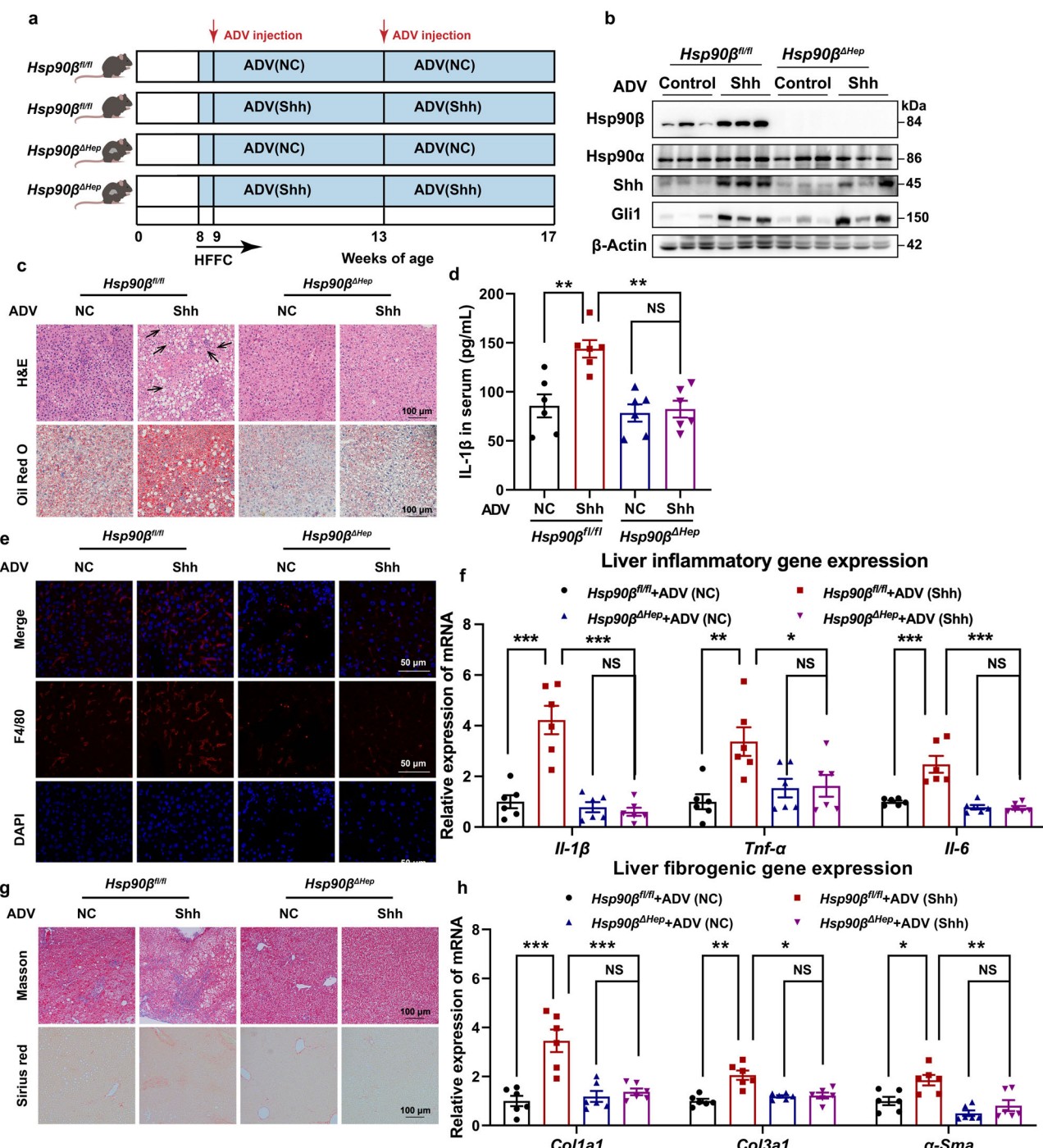

**Fig. 3 | Hepatic deletion of *Hsp90β* blocks the facilitation of Shh on the NASH process. a** Experimental scheme of *Hsp90β^fl/fl* or *Hsp90β^ΔHep* mice injected with ADV–NC or ADV-Shh on HFFC diet. **b** Western blot analysis in the liver tissues of *Hsp90β^fl/fl* and *Hsp90β^ΔHep* mice injected with ADV–NC or ADV-Shh on HFFC diet (*n* = 6 mice per group). **c** Representative H&E-stained FFPE liver sections and ORO-stained frozen liver sections. Each arrow indicated hepatocyte ballooning. **d** Serum IL-1β levels of *Hsp90β^fl/fl* and *Hsp90β^ΔHep* mice injected with ADV–NC or ADV-Shh on HFFC diet (*Hsp90β^fl/fl*–NC versus *Hsp90β^fl/fl*-Shh, *P* = 0.0018, *Hsp90β^fl/fl*-Shh versus *Hsp90β^ΔHep*-Shh, *P* = 0.0010, *Hsp90β^ΔHep*-NC versus *Hsp90β^ΔHep*-Shh, *P* = 0.99). **e** Representative liver sections stained with F4/80. **f** The expression of inflammatory genes in livers of *Hsp90β^fl/fl* and *Hsp90β^ΔHep* mice injected with ADV–NC or ADV-Shh on HFFC diet (for *Il-1β*, *Hsp90β^fl/fl*–NC versus *Hsp90β^fl/fl*–Shh, *P* = 3.6E-06, *Hsp90β^fl/fl*-Shh versus *Hsp90β^ΔHep*-Shh, *P* = 6.7E-07, *Hsp90β^ΔHep*-NC versus *Hsp90β^ΔHep*-Shh, *P* = 0.96; for *Tnf-α*, *Hsp90β^fl/fl*–NC versus *Hsp90β^fl/fl*–Shh, *P* = 0.0022, *Hsp90β^fl/fl*-Shh versus *Hsp90β^ΔHep*-Shh, *P* = 0.023, *Hsp90β^ΔHep*-NC versus

*Hsp90β^ΔHep*-Shh, *P* = 0.99; for *Il-6*, *Hsp90β^fl/fl*–NC versus *Hsp90β^fl/fl*–Shh, *P* = 2.8E-05, *Hsp90β^fl/fl*-Shh versus *Hsp90β^ΔHep*-Shh, *P* = 3.5E-06, *Hsp90β^ΔHep*-NC versus *Hsp90β^ΔHep*-Shh, *P* = 0.99). **g** Representative liver sections stained with Masson or Sirius red. **h** The expression of fibrogenic genes in livers of *Hsp90β^fl/fl* and *Hsp90β^ΔHep* mice injected with ADV–NC or ADV-Shh on HFFC diet (for *Col1a1*, *Hsp90β^fl/fl*–NC versus *Hsp90β^fl/fl*–Shh, *P* = 1.6E-05, *Hsp90β^fl/fl*-Shh versus *Hsp90β^ΔHep*-Shh, *P* = 1.3E-04 *Hsp90β^ΔHep*-NC versus *Hsp90β^ΔHep*-Shh, *P* = 0.94; for *Col3a1*, *Hsp90β^fl/fl*-NC versus *Hsp90β^fl/fl*–Shh, *P* = 1.8E-05, *Hsp90β^fl/fl*-Shh versus *Hsp90β^ΔHep*-Shh, *P* = 3.1E-04, *Hsp90β^ΔHep*-NC versus *Hsp90β^ΔHep*-Shh, *P* = 0.99; for *α-Sma*, *Hsp90β^fl/fl*-NC versus *Hsp90β^fl/fl*–Shh, *P* = 0.011, *Hsp90β^fl/fl*-Shh versus *Hsp90β^ΔHep*-Shh, *P* = 0.0023, *Hsp90β^ΔHep*-NC versus *Hsp90β^ΔHep*-Shh, *P* = 0.99). Data are presented as mean ± SEM. *n* = 6 mice per group. \**P* < 0.05, \*\**P* < 0.01, \*\*\**P* < 0.001, NS no significant difference, one-way ANOVA. Source data are provided in the Source Data file.

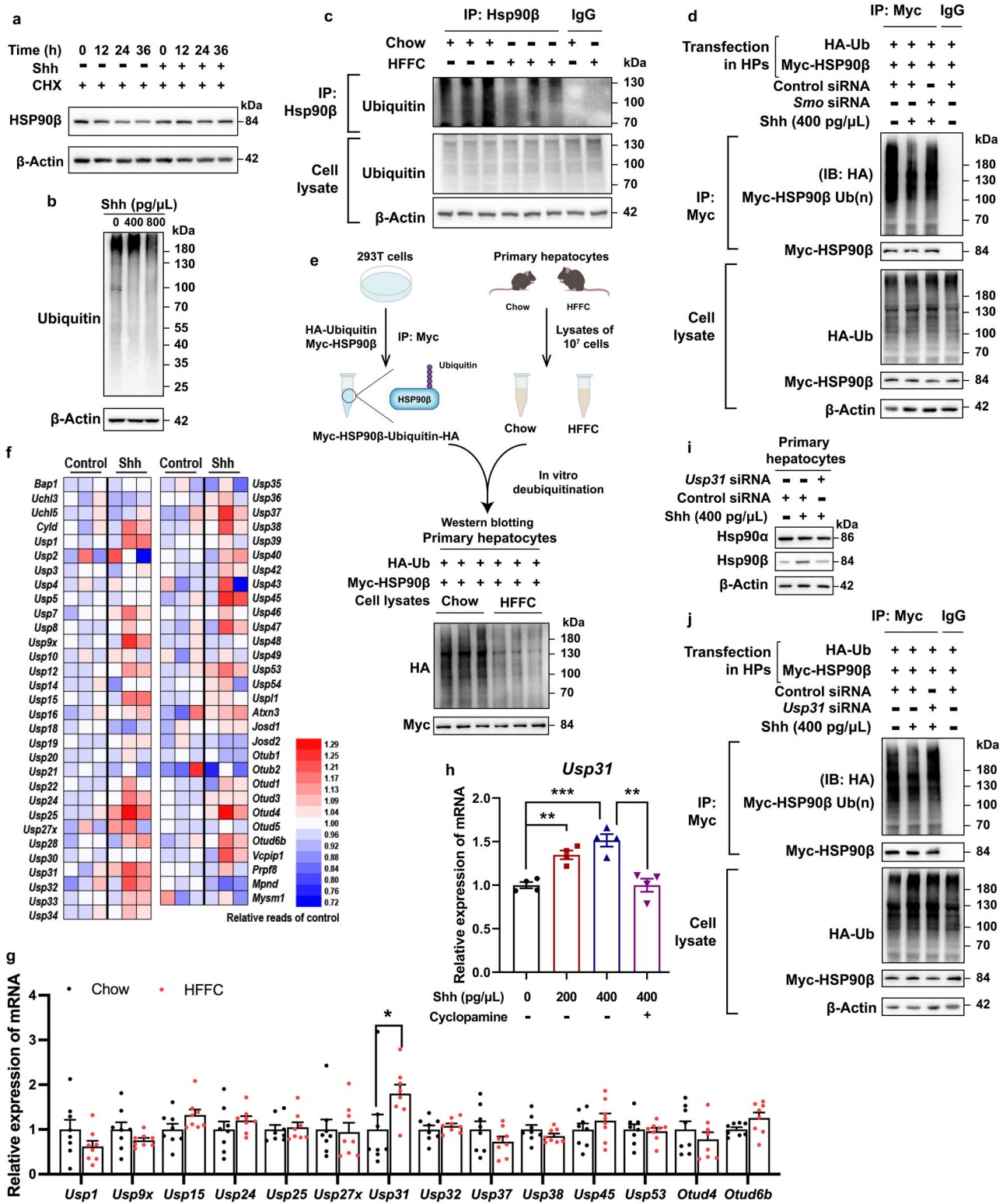

distribution (Fig. 5f, g). Interestingly, the knockdown of *Gli1* did not affect Shh-induced *Usp31* expression. In comparison, *Smo* knockdown resulted in the blockade of *Usp31* upregulation (Fig. 5h). These results suggested that Bmal1 and Gli1 behaved completely differently in regulating the expression of *Usp31*. Shh promoted the dissociation of Bmal1 and Sufu, thus facilitating the nuclear translocation of Gli1 to transcribe its target genes[17]. Similarly, Shh also enabled the dissociation of Bmal1 from Sufu and increased Bmal1's nuclear distribution (Fig. 5i, j).

## Shh induces inflammation via exosomes

Although Shh drove the pathogenesis of NASH including steatosis, inflammation, and fibrosis, it did not induce the expressions of *Il-1β*, *Tnf-α*, and *Il6* in bone marrow-derived macrophages (BMDMs) (Fig. 6a). We then sought to unravel the underlying mechanisms. Hsp90β is a key factor that regulates the release of exosomes[12]. Therefore, we investigated whether Shh-Hsp90β affected exosomal secretion in HFFC diet-fed mice. Liver exosomes from mice were successfully obtained with a diameter between 50 and 100 nm, enriched in the

**Fig. 4 | Shh promotes Hsp90β deubiquitylation through Usp31. a** Hepa 1–6 cells were treated with CHX, then exposed to 400 pg/μL Shh for 0–36 h. The cell lysates were analyzed by western blot. **b** Hepa 1–6 cells were treated with 400 or 800 pg/μL Shh for 24 h, protein ubiquitylations were analyzed by western blot. **c** HSP90β ubiquitylation assays were performed in primary hepatocytes isolated from mice fed with chow/HFFC diet for 16 weeks and analyzed by western blot. The cell lysates were incubated with immunoprecipitated HSP90β and then immunoblotted with anti-ubiquitin to reveal the ubiquitylation levels. **d** HSP90β ubiquitylation assays were performed in primary hepatocytes treated with 400 pg/μL Shh after Smo knocked down and plasmid Myc-HSP90β and HA-ubiquitin transfection. The cell lysates were incubated with immunoprecipitated Myc and then immunoblotted with anti-HA to reveal the ubiquitylation levels. **e** Schematic representation of in vitro deubiquitylation assay. The ubiquitylated Myc-HSP90β was immunoprecipitated from HEK293T cells, incubated with the lysates from primary hepatocytes isolated from mice fed with chow/HFFC diet for 16 weeks. **f** The expression heat-map of DUB genes in Hepa 1–6 cells treated with or without Shh (*n* = 3 mice per group). **g** The expression of 14 DUB genes in the livers of mice on a normal chow or HFFC diet for 25 weeks (for *Usp1*, chow versus HFFC, *P* = 0.68, for *Usp9x*, chow versus HFFC, *P* = 0.99, for *Usp15*, chow versus HFFC, *P* = 0.87, for *Usp24*, chow versus HFFC, *P* = 0.99, for *Usp25*, chow versus HFFC, *P* = 0.99, for *Usp27x*, chow versus HFFC, *P* = 0.99, for *Usp31*, chow versus HFFC, *P* = 0.034, for *Usp32*, chow versus HFFC, *P* = 0.99, for *Usp37*, chow versus HFFC, *P* = 0.96, for *Usp38*, chow versus HFFC, *P* = 0.99, for *Usp45*, chow versus HFFC, *P* = 0.99, for *Usp53*, chow versus HFFC, *P* = 0.99, for *Otud4*, chow versus HFFC, *P* = 0.99, for *Otud6b*, chow versus HFFC, *P* = 0.97). **h** *Usp31* gene expression in Hepa 1–6 cells treated 0–400 pg/μL Shh with or without 20 μM cyclopamine for 24 h (Shh-0 versus Shh-200, *P* = 0.0040, Shh-0 versus Shh-200, *P* = 0.0002, Shh-400 versus Shh-400+ cyclopamine, *P* = 0.0025). **i** Primary hepatocytes were transfectied with *Usp31* RNAi for 48 h, then cells were treated with 400 pg/μL Shh for 24 h, Hsp90β protein level was analyzed by immunoblotting. **j** HSP90β ubiquitylation assays were performed in primary hepatocytes treated with 400 pg/μL Shh after Usp31 knocked down and plasmid Myc-HSP90βand HA-ubiquitin transfection. The cell lysates were incubated with immunoprecipitated Myc and then immunoblotted with anti-HA to reveal the ubiquitylation levels. Data are presented as mean ± SEM. *n* = 6 mice per group. \*P < 0.05, \*\*P < 0.01, \*\*\*P < 0.001, NS no significant difference, one-way ANOVA. Source data are provided in the Source Data file.

classical exosomal markers, Tsg101 and Cd63, and depleted in cellular markers, including calnexin (Supplementary Fig. 10a–c). There was a clear increase in the number of liver exosomes from HFFC diet-fed mice (Fig. 6b). Liver exosomes from HFFC diet-fed mice exhibited significant pro-inflammatory effects (Fig. 6c). In contrast, hepatocellular *Hsp90β* ablation fully abrogated the increase in liver exosomes, as well as inflammatory cytokine expression in BMDMs (Fig. 6b, c). Taken together, these data indicated that Shh-induced NASH development required Hsp90β-facilitated extracellular communication through exosomes.

### miRNAs are the mediators promoting macrophage inflammation

miRNAs stored and released in exosomes and other small extracellular vesicles (EVs) elicit changes in gene expression and mediate the intercellular communications[41]. To investigate the specific features of miRNAs in exosomal release, exosomes were isolated from chow diet-, HFFC diet-fed *Hsp90β^fl/fl^*, or *Hsp90β^ΔHep^* mice. The miRNA composition of the secreted exosomes was analyzed by sequencing. Of the 722 miRNAs in the liver exosomes, 11 miRNAs were significantly highly expressed in HFFC diet-fed mice, while that of 8 miRNAs reversed after hepatocellular knocking out *Hsp90β* (Fig. 6d). The miRNAs contained in the intersection of the two were miR-28-5p, miR-3473a, miR489-3p, and miR-500-3p. These miRNAs were mediated by Hsp90β and increased in NASH (Fig. 6d).

To further validate which miRNAs played a key role in regulating inflammatory responses, BMDMs were incubated with corresponding miRNA mimics of miR-28-5p, miR-3473a, miR489-3p, and miR-500-3p. miR-28-5p and miR-3473a induced *Il-1β*, *Tnf-α*, and *Il6* expression, while miR489-3p only significantly promoted *Il6* expression (Fig. 6e). Because miR-28-5p is conserved between rodents and humans, while miR-3473a is a mouse-specific miRNA, we focused on the role of miR-28-5p in NASH. We found that Shh promoted the expression of miR-28-5p in exosomes secreted by primary hepatocytes of *Hsp90β^fl/fl^* mice, which was reversed following Hsp90β ablation (Fig. 6f). Further, sera from healthy volunteers and NASH patients were collected. As expected, circulating exosomes were significantly elevated in NASH patients (Fig. 6g). The abundance of serum miR-28-5p increased markedly in NASH patients (Fig. 6h), while serum SHH elevated without statistical significance (Fig. 6i). Similarly, Shh promoted inflammatory gene expression in organoid derived from *Hsp90β^fl/fl^* mice. Increased *Il-1β*, *Tnf-α* and *Il-6* expression could be blunted either in hepatic organoid derived from liver-specific *Hsp90* knockout mice (*Hsp90β^ΔHep^*), or in hepatic organoid derived from *Hsp90β^fl/fl^* mice treated with miR-28-5p antagomir (Supplementary Fig. 11a–c). miR-28-5p also promoted fibrogenic genes expression in HSC and organoids when administered with miR-NC mimic and miR-28-5p mimic (Supplementary Fig. 11d, e). Taken together, these results indicated that Shh-induced NASH development required HSP90β-facilitated extracellular communications through miR-28-5p.

### miR-28-5p target *Rap1b* to stimulate the expression of inflammatory cytokines via NF-κB

miRNAs regulate gene expression by binding to their seed sequence, thus targeting mRNAs and altering their translation efficiency and/or stability[42]. To identify the targets of miR-28-5p, miRDB[43] and TargetScan[44] were used to predict a set of common target genes in *Homo sapiens* and *Mus musculus*. Among these, 7 predicted genes were chosen out because they were predicted as miR-28-5p both in *Homo sapiens* and *Mus musculus* (Fig. 7a). Among these, 7 predicted genes were chosen because they were predicted for miR-28-5p both in *Homo sapiens* and *Mus musculus* (Fig. 7a). Among them, the antibodies against Slc44a5 and Gpm6a did not show any signals in BMDM lysates. After the exogenous treatment with miR-28-5p mimics, only RAS-related protein 1b (Rap1b) expression was downregulated (Fig. 7b). Subsequently, wild-type and mutated miR-28-5p-binding sites in the 3′-UTR of the *Rap1b* gene were cloned into a pmirGlo luciferase reporter. In the presence of miR-28-5p, BMDMs transfected with pmirGlo carrying the wild-type binding site exhibited a marked decrease in luciferase activity (Fig. 7c). In comparison, the pmirGlo carrying a mutated binding site did not cause such repression in activity (Fig. 7c). Consistent with previous studies, miR-28-5p suppressed Rap1b, leading to the activation of Ras activity, thereby upregulating the Pi3k-Akt signaling pathway (Fig. 7d). Pi3k-Akt subsequently phosphorylated Ikk and IκB, causing IκB degradation and promoting the nuclear translocation of p65 (Fig. 7e, f)[45].

### NPs carrying miR-28-5p antagomir attenuate inflammation in HFFC-diet-induced NASH mice

Since miR-28-5p antagomir treatment reversed inflammation in BMDMs (Fig. 7d, e), we then conducted in vivo studies in experimental NASH mice. RNAi therapeutics, like small interfering RNAs (siRNAs) or miRNAs, have significant advantages of high specificity and efficiency over conventional chemotherapeutics. Recently, two siRNA drugs, Onpattro (patisiran) and Givlaari (givosiran) were approved by the FDA (U.S. Food and Drug Administration). These are used for the treatment of peripheral nerve disease (polyneuropathy) and acute hepatic porphyria, respectively. Ternary nanocomplex (NC) composed of miR-NC or miR-28-5p antagomir, a synthetic thiol-bearing methacrylated hyaluronic acid (sm-HA), and protamine formed through an electrostatic-driven physical assembly, were chemically crosslinked to concurrently acquire the collaboratively

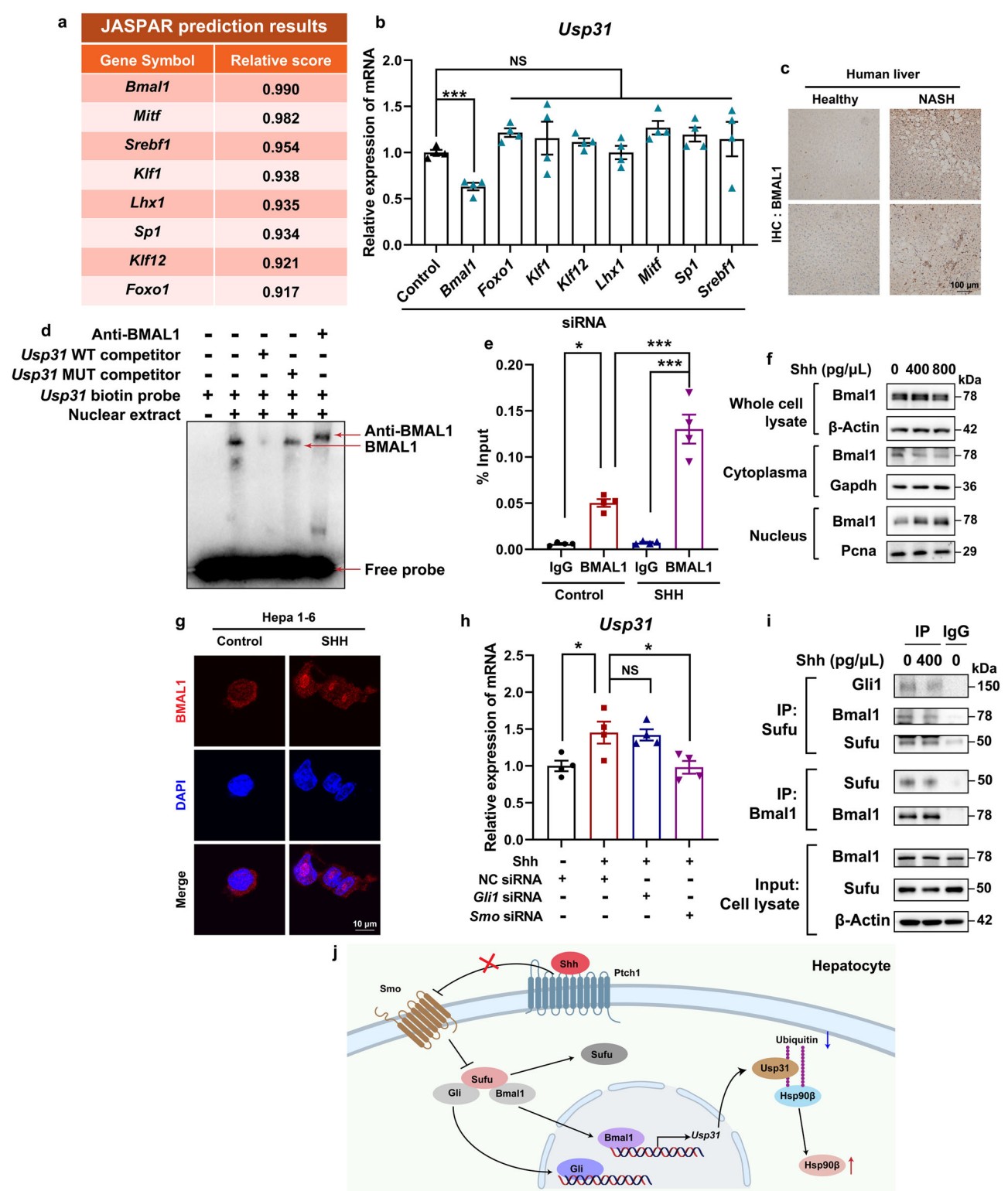

assembled nanocapsule (cNC). The miRNA-packing capacity was evaluated by the agarose gel electrophoretic assay (Supplementary Fig. 12a). Compared to the same amount of free miR-28-5p antagomir, the encapsulation efficiency was the best at N/P of 40. The particle size of nanoparticles (NPs) was characterized to be approximately 100 nm (Supplementary Fig. 12b). The in vivo targeting capacity of NP-containing Cy5-miR-28-5p antagomir was monitored using an in vivo imaging system (IVIS). Both at 1 h and 24 h after injection, NPs were mainly accumulated in the liver (Supplementary Fig. 12c). Mice were fed

chow or HFFC diet for 8 weeks followed by treatment with NPs containing miR-NC or miR-28-5p antagomir (5 nmol/mouse, twice a week) (Fig. 8a). In miR-NC treated mice, the HFFC diet promoted hepatic steatohepatitis, including lipid accumulation in the liver (Fig. 8b), along with an increase in the serum TC and TG levels (Fig. 8c, d), hepatic damage (Fig. 8e, f), and inflammation (Fig. 8g–k). Administration of miR-28-5p antagomir did not influence the hepatic lipid content in mice while mitigating the HFFC-induced hepatic damage and inflammation which suppressed the development of NASH (Fig. 8b–k).

**Fig. 5 | BMAL1 is the transcription factors of *Usp31*. a** The transcription factors (TFs) of *Usp31* were predicted using JASPAR. **b** Each of the predicted *Usp31* TFs was knocked down by corresponding siRNA, the expression of *Usp31* was then analyzed by qRT-PCR (Control siRNA versus *Bmal1* siRNA, $P = 0.0003$, versus *Foxo1* siRNA, $P = 0.53$, versus *Klf1* siRNA, $P = 0.82$, versus *Klf12* siRNA, $P = 0.956$, versus *Lhx1* siRNA, $P = 0.99$, versus *Mitf* siRNA, $P = 0.31$, versus *Sp1* siRNA, $P = 0.64$, versus *Srebf1* siRNA, $P = 0.86$). **c** Representative immunohistochemistry staining of BMAL1 in the livers of healthy donors and NAFLD patients. **d** The EMSA assays were carried out with nuclear fraction of Hepa 1–6 cells. **e** Hepa 1–6 cells were treated with 400 pg/ μL Shh for 24 h, the binding of BMAL1 to the *Usp31* promoter region was analyzed by ChIP analysis (Control+IgG versus Control+BMAL1, $P = 0.001$, Control+BMAL1 versus SHH + BMAL1, $P = 4.3E-05$, SHH+IgG versus SHH + BMAL1, $P = 5.1E-07$). **f** Hepa 1–6 cells were treated with 400 pg/μL Shh for 24 h, cytoplasmic and nuclear

BMAL1 were analyzed by western blot. **g** Hepa 1–6 cells were treated with 400 pg/ μL Shh for 24 h, the cellular Bmal1 distribution was analyzed by immuno-fluorescence staining. **h** Hepa 1–6 cells were treated indicated siRNAs for 48 h, and then incubated with 400 pg/μL Shh for 24 h, the gene expression of *Usp31* was analyzed by qRT-PCR (Shh+NC siRNA versus NC siRNA, $P = 0.020$, versus Shh+*Gli1* siRNA, $P = 0.99$, versus Shh+*Smo* siRNA, $P = 0.016$). **i** Hepa 1–6 cells were treated with 400 pg/μL Shh for 24 h, the interaction between Bmal1/Gli1 and Sufu was analyzed by immunoprecipitation and western blot. **j** The signaling pathway through which Shh promotes Hsp90β deubiquitylation by Usp31. Data are presented as mean ± SEM. $n = 6$ mice per group. *$P < 0.05$, **$P < 0.01$, ***$P < 0.001$, NS no significant difference, one-way ANOVA. Source data are provided in the Source Data file.

## Discussion

The correlation between SHH and NAFLD has been previously suggested. For instance, Shh stimulates the expression of *Hilnc*, which directly interacts with IGF2BP2, thus enhancing *Pparγ* mRNA stability[31]. Most of these studies suggest a critical role of Gli1, the major transcriptional factor in Shh signaling. In the noncanonical HH signaling pathway, SMO-independent stimulation of GLI activity relies on transcriptional or post-translational regulation of the Gli protein[46]. In our study, we identified a Gli1-independent role of Shh signaling. The activation of the SHH pathway promoted Hsp90β stability by increasing the expression of *Usp31* via Bmal1 (Figs. 4 and 5). Mechanistically, this was similar to SMO-dependent Gli1 activation which includes dissociation from Sufu (Fig. 5i). Bmal1 (brain and muscle ARNT-like 1) and Clock (circadian locomotor output cycles kaput) are two essential components of the circadian clock[47]. Circadian misalignment has been identified as a risk factor for metabolic disease[48]. The circadian clock is involved in regulation of hepatic triglyceride accumulation, inflammation, oxidative stress, and mitochondrial dysfunction[49–51]. The activity of BMAL1 positively correlates with liver fibrosis, and lipid accumulation decreases in *Bmal1* KO mice, which likely explains the reduced inflammation and fibrosis in these mice[52]. Additionally, silencing *BMAL1* prevents the recovery of *FAS* and *ACC* gene expression, therefore attenuating lipogenesis in HepG2 cells. In this study, we found a regulatory mode of Bmal1 activity by Shh. Activation of the SHH pathway promoted Bmal1's nuclear translocation, thus initiating *Usp31* transcription. It would be interesting to investigate whether Shh affects circadian rhythm. If *Usp31* is also transcribed by the Bmal1/Clock complex, it might contribute to shaping the circadian ubiquitylated proteome[53].

HSP90 facilitates the function of numerous proteins in cancer cells, and is often used as a loading control in WB assay. Evidence showing paralog-specific roles of HSP90β in diseases is emerging[9,37]. We found that HSP90β was positively correlated with disease progression, while the expression of HSP90α was not related to disease progression (Fig. 2m–o). However, the limitation of this study was that the sample size was inadequate to demonstrate a significant difference between Hsp90β in NAFL and NASH patients ($p = 0.0828$).

In this study, the clinical sample size was not large enough, and the p value was not within 0.05. More samples are needed to support the clinical correlation.

Using *Hsp90β*$^{ΔHep}$ mice, we provided genetic evidence for the contribution of Hsp90β contributed to NASH development. Post-translational modifications (PTM) of HSP90 are crucial to its various biological processes. HDAC6, a deacetylase of HSP90, impairs the chaperone function of HSP90 by decreasing its affinity to ATP and client proteins. The inhibition of serine/threonine phosphatase results in hyperphosphorylation of HSP90 and compromised chaperoning of the classic kinase client v-Src. Interestingly, HECT domain E3 ubiquitin-protein ligase (HECTD1) and carboxyl terminus of Hsc70-interacting protein (CHIP) participate as E3 ligases for HSP90's ubiquitylation and degradation. Ubiquitylation and deubiquitylation are counter-

balancing PTMs. In this study, we identified the DUB of HSP90β (Fig. 4). This finding may facilitate the assessment of HSP90β's stability, activity, and localization under different physiological and pathological conditions.

Exosomes, a type of NP, are secreted by all cell types and serve as a vehicle for intercellular and intra-organ communication. Their role in regulating tissue and intercellular metabolic signaling is increasingly gaining traction. In both NASH patients and mouse models of NASH, circulating exosomes observably increase (Fig. 6b, h). In a previous study, when hepatocytes from lean/healthy human or chow diet-fed mice were treated with palmitic acid (PA, 500 μM), exosomes increased markedly[54]. RAB27, a critical protein for exosomal secretion, is upregulated in hepatocytes treated with PA or in hepatocytes from human NAFLD/NASH patients[54]. Moreover, in hepatocyte-specific *Rab27* KO mice, iron in EVs decreased significantly, therefore attenuating fibrosis in a mouse model of NASH. HSP90 plays an important role in regulating exosomal secretion. Unsurprisingly, in *Hsp90β*$^{ΔHep}$ mice, HFFC diet or Shh-stimulated EV secretion was largely abrogated (Fig. 6b).

B-scan ultrasonography, computed tomography (CT) scan, and Fibroscan are used to identify steatosis, fibrosis, and cirrhosis with reliable accuracy[55]. However, these methods cannot be used for the diagnosis of NASH, an advanced stage of NAFLD. Although the patient's compliance with tissue biopsy is poor, an accurate diagnosis of NASH is necessary. Thus, it is particularly important to develop noninvasive diagnostic methods for NASH. Many miRNAs have been confirmed as diagnostic markers for NAFLD or NASH[56,57] but if these also serve as therapeutic targets remain unknown. Herein, we showed that the levels of miR-28-5p increased significantly in a mouse model of NASH and NASH patients (Fig. 6d, g), the inhibitor of miR-28-5p successfully abrogated macrophage inflammation in vitro and in vivo (Fig. 8). miR-28-5p is a tumor suppressor with several targets[58–62]. Consistent with some previous reports[60,63–66], we also identified Rap1b as the target of miR-28-5p (Fig. 7a–c). Although miR-28-5p may regulate SREBPs[67], its role in metabolic diseases has never been studied. Our findings suggested that miR-28-5p could be used as a diagnostic marker and a therapeutic target for NASH. At the same time, we also tested the concentration of SHH in the serum with an ELISA kit. Although the average concentration of SHH in the serum of NASH patients was higher than healthy controls, but there was no significant difference between the two groups. In healthy control group, SHH concentration of 4 volunteers was far higher than others in the same group. SHH secretion elevated not only in NASH but also in pulmonary fibrosis[68], chronic kidney disease[69], and alcohol-associated liver injury[70]. Therefore, we think that miR-28-5p in exosomes from serum serves as a better marker to predict the severity of NASH patients.

The contributions of macrophages to the development of NASH have been extensively studied. M1/M2 polarization of macrophages is an important step in liver inflammation, liver cell damage, and satellite cell activation[71]. However, our knowledge of pathways regulating macrophage polarization is scarce[72]. Hepatocyte-originating signals,

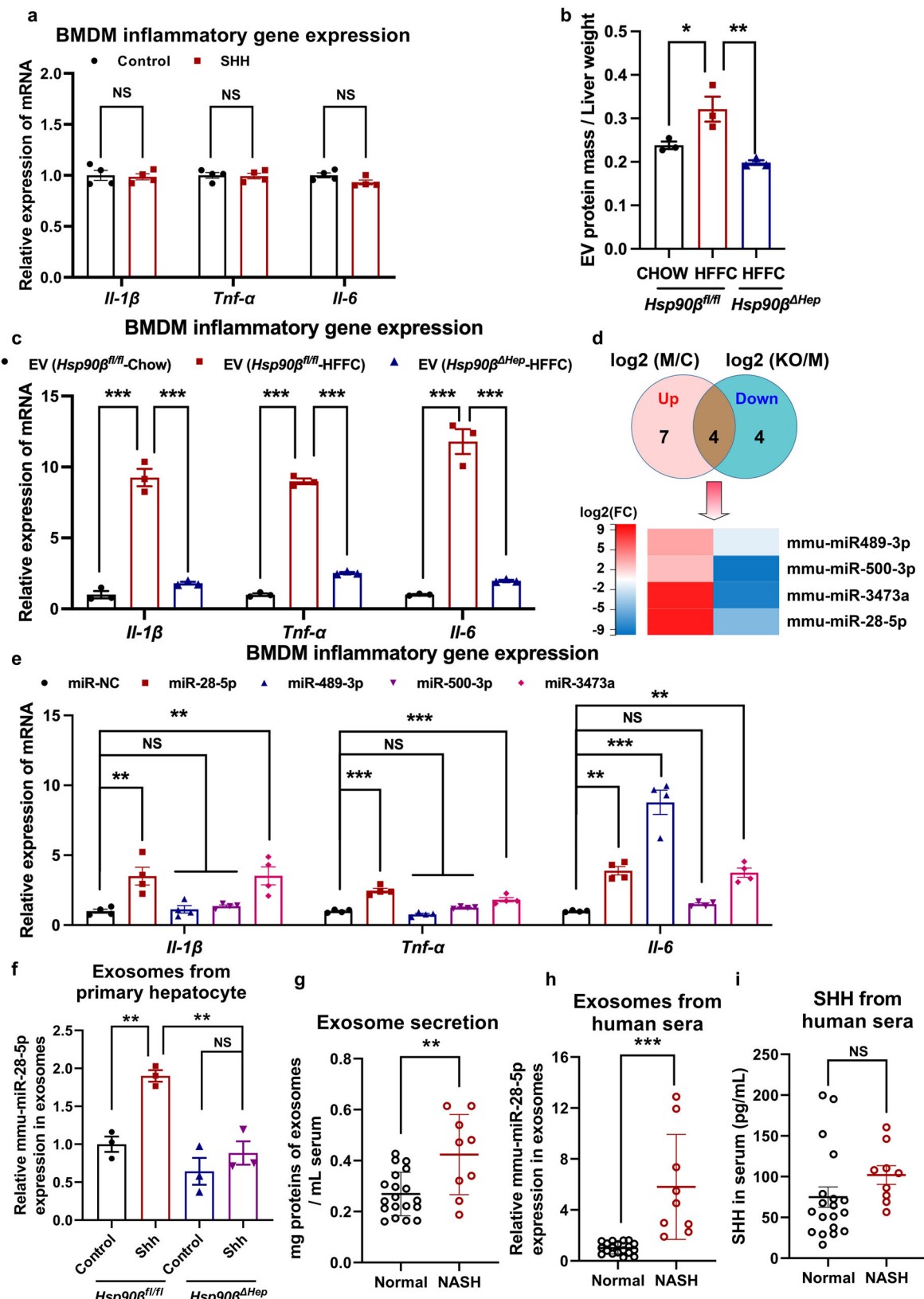

including soluble chemokines like C-C motif chemokine ligand 2/ monocyte chemoattractant protein 1 (CCL2/MCP1) and the receptors C-C motif chemokine receptor (CCR) 2 and 5 contribute to the recruitment of proinflammatory macrophages[73,74]. EVs are a type of hepatocyte-derived mediator of intercellular communication[75]. EVs recruit monocyte-derived macrophages to the liver, resulting in injury

and inflammation in diet-induced steatohepatitis. In hepatocytes-derived EVs, let-7e-5p and miR-210-3p target adipocytes, thus promoting lipid accumulation. miR-192-5p induces the activation of pro-inflammatory macrophages. miR-128-3p suppresses fibrotic gene expression in HSCs. In this study, we found that hepatocyte-secreted miR-28-5p aggravated inflammation by decreasing Rap1b and

**Fig. 6 | Shh induces inflammation via miR-28-5p in exosomes. a** BMDMs were treated with 400 pg/μL Shh for 24 h, the expression of inflammatory genes was analyzed by qRT-PCR. $n = 4$ independent experiments per group. **b** Relative exosome amount was analyzed in the liver of *Hsp90β*$^{fl/fl}$ or *Hsp90β*$^{ΔHep}$ mice fed with chow or HFFC diet. ($n = 3$ mice per group, chow+*Hsp90β*$^{fl/fl}$ versus HFFC+*Hsp90β*$^{fl/fl}$, $P = 0.027$, HFFC +*Hsp90β*$^{fl/fl}$ versus HFFC+ *Hsp90β*$^{ΔHep}$, $P = 0.0046$). **c** BMDMs were treated with exosomes derived from livers in *Hsp90β*$^{fl/fl}$ and *Hsp90β*$^{ΔHep}$ mice fed with chow or HFFC for 24 h, the expression of inflammatory genes was analyzed by qRT-PCR ($n = 3$ mice per group, for *Il-1β*, chow+*Hsp90β*$^{fl/fl}$ versus HFFC+*Hsp90β*$^{fl/fl}$, $P = 9.7E-06$, HFFC +*Hsp90β*$^{fl/fl}$ versus HFFC+ *Hsp90β*$^{ΔHep}$, $P = 1.8E-05$; for *Tnf-α*, chow +*Hsp90β*$^{fl/fl}$ versus HFFC+*Hsp90β*$^{fl/fl}$, $P = 4.8E-08$, HFFC +*Hsp90β*$^{fl/fl}$ versus HFFC+ *Hsp90β*$^{ΔHep}$, $P = 1.0E-07$; for *Il-6*, chow+*Hsp90β*$^{fl/fl}$ versus HFFC+*Hsp90β*$^{fl/fl}$, $P = 1.0E-05$, HFFC +*Hsp90β*$^{fl/fl}$ versus HFFC+ *Hsp90β*$^{ΔHep}$, $P = 1.7E-05$). **d** Exosomes were collected from livers of *Hsp90β*$^{fl/fl}$ and *Hsp90β*$^{ΔHep}$ mice fed with chow or HFFC diet, the expression of exosomes containing miRNA were analyzed by sequencing, the heatmap of miRNA expression was shown. **c** represented *Hsp90β*$^{fl/fl}$ mice on a normal chow diet. M represented *Hsp90β*$^{fl/fl}$ mice on a HFFC diet. KO represented *Hsp90β*$^{ΔHep}$ mice on a HFFC diet. $n = 3$ mice per group. **e** BMDM cells were administrated with indicated miRNAs for 24 h, the expression of inflammatory genes was analyzed by qRT-PCR ($n = 4$ independent experiments per group, for *Il-1β*, miR-NC

versus miR-28-5p, $P = 0.0031$, versus miR-489-3p, $P = 0.99$, versus miR-500-3p, $P = 0.93$, versus miR-3473a, $P = 0.0029$; for *Tnf-α*, miR-NC versus miR-28-5p, $P = 4.3E-07$, versus miR-489-3p, $P = 0.42$, versus miR-500-3p, $P = 0.33$, versus miR-3473a, $P = 3.5E-04$; for *Il-6*, miR-NC versus miR-28-5p, $P = 0.0011$, versus miR-489-3p, $P = 4.1E-08$, versus miR-500-3p, $P = 0.84$, versus miR-3473a, $P = 0.0017$). **f** Exosomes were collected from primary hepatocytes of *Hsp90β*$^{fl/fl}$ and *Hsp90β*$^{ΔHep}$ mice after the treatment with 400 pg/μL Shh for 48 h, and relative expression of miR-28-5p was analyzed by qRT-PCR ($n = 3$ mice per group, control+*Hsp90β*$^{fl/fl}$ versus Shh+*Hsp90β*$^{fl/fl}$, $P = 0.0061$, Shh+*Hsp90β*$^{fl/fl}$ versus Shh+*Hsp90β*$^{ΔHep}$, $P = 0.0029$, control+*Hsp90β*$^{ΔHep}$ versus Shh+*Hsp90β*$^{ΔHep}$, $P = 0.60$). **g** Exosomes were collected from sera of healthy donors or NASH patients, and the relative exosome amount was analyzed ($n = 19$ participants in normal group. $n = 9$ participants in NASH group, normal versus NASH, $P = 0.0022$). **h** Exosomes were collected from sera of healthy donors or NASH patients, and relative expression of miR-28-5p was analyzed by qRT-PCR ($n = 19$ participants in normal group. $n = 9$ participants in NASH group, normal versus NASH, $P = 2.4E-05$). **i** Serum SHH concentration of healthy donors or NASH patients ($n = 19$ participants in normal group. $n = 9$ participants in NASH group, $P = 0.19$). Data are presented as mean ± SEM. *$P < 0.05$, **$P < 0.01$, ***$P < 0.001$, NS no significant difference, one-way ANOVA. Source data are provided in the Source Data file.

inhibiting AKT-mediated IKK activation, thus preventing NF-κB activation in macrophages (Fig. 7d, e). These results expand our understanding of hepatocyte-originating mediator-regulating macrophage polarization during NASH development.

The mechanism of SHH pathway is complicated (including negative feedback), and its role in NASH still requires lots of efforts to uncover. We find that Shh increases Hsp90β protein levels in hepatocytes, while studies have showed that Hsp90β promotes de novo lipid synthesis in hepatocytes by increasing the expression of SREBP[9,76]. However, Matz-Soja reveals that conditional hepatocyte-specific deletion of Smo promoted a GLI-code associated steatosis[77]. In this research, 8-weeks old mice are fed with normal diet for 5 weeks before sacrifice, in which case SHH signaling is inactive. Another research demonstrates that Shh treatment in 3T3-L1 cells prevents adipogenesis and SHH pathway is inhibited in adipose tissue of HFD-induced obese mice[78]. These conclusions differed from the phenotypes we discovered in mouse NASH models. It should be noted that in these reports, data were collected in the condition that SMO activity is repressed by PTCH1, and its downstream pathways could not be inhibited any more.

In summary, HSP90β is an important downstream target of Shh that is required for Shh-stimulated NASH development. This process also includes changes in HSP90β-mediated hepatocyte exosome secretion. The exosomes containing miRNAs (like miR-28-5p and miR-3473a) mediated macrophage polarization and stimulated the expression of inflammatory cytokines. Our data raised the possibility of using serum miR-28-5p levels as a non-invasive diagnostic marker for NASH. NPs carrying an antagomir directed against miR-28-5p might also exert potential beneficial effects on experimental NASH models.

## Methods
### Reagents
Key reagents, antibodies, and primers used in this study are listed in Supplemental Materials.

### Ethics statements
All experimental protocols involving human participants (human liver sections and human blood samples) have been performed in accordance with the ICH Guidelines for Good Clinical Practice. As written in their respective datasheets, ethical approvals have been obtained by the relevant parties and all participants gave written informed consent. The collection and use of human liver and blood samples for this study were approved by the Ethical Committee of Guangdong Provincial People's Hospital (Guangdong Academy of Medical Sciences) and the Beijing You'An Hospital. Animal experiments were conducted following the

Guidelines for Animal Experimentation of China Pharmaceutical University, and the protocols were approved by the Science and Technology Department of Jiangsu Province (SYXK (SU) 2016-0011). The reporting of animal experiments complies with the ARRIVE guidelines.

### Mice
Adult 6-week-old C57BL/6 male mice weighing about 20 g, were purchased from Beijing Vital River Laboratory Animal Technology (Beijing, China) and housed in a specific-pathogen-free facility. All mice used herein were maintained in a temperature- (22 ± 2 °C) and humidity-controlled (40–70%) animal facility under a 12 h/12 h light/dark cycle and allowed free access to food and water. As shown in Fig. 1b, the mice were fed a high fat, high fructose, and high cholesterol diet (HFFC containing 40 kcal% fat (mostly palm oil), 20 kcal% fructose, and 2% cholesterol; #D09100310, Research Diets) or a normal chow diet for 16 weeks with *ad libitum* access. Subsequently, vehicle or cyclopamine (20 mg/kg) was intraperitoneally injected every 2 days for 9 weeks as indicated.

To evaluate the effects of 17AAG, mice were fed HFFC or a normal chow diet *ad libitum* for 16 weeks (Supplementary Fig. 5a), following which, vehicle or 17AAG (1 mg/kg) was every 2 days for 9 weeks to the respective groups.

*Hsp90β*-flox mice were generated by Viewsolid Biotech using the CRISPR-cas9 system. sgRNA-directed cas9 endonuclease cleavage inserted LoxP sites at both ends of exons 1-3 by homologous recombination (Supplementary Fig. 2a). *Hsp90β*-flox mice were then mated with *Albumin*-Cre mice to produce the *Hsp90β*$^{fl/fl}$; *Alb-cre*$^{+/wt}$ (*Hsp90β*$^{ΔHep}$) mice (Supplementary Fig. 2a–c). The 8-weeks old *Hsp90β*$^{fl/fl}$ or *Hsp90β*$^{ΔHep}$ mice were fed chow or HFFC diet for 33 weeks. At the 9th week, mice were injected with ADV-NC or ADV-Shh $1.0 × 10^9$ pfu per mouse through their tail veins and fed HFFC diet for an additional 9 weeks (Fig. 3a).

To evaluate the therapeutic effects of miRNA inhibitors, mice were fed chow or HFFC diet for 8 weeks. Subsequently, nanoparticles encapsulating miR-NC or miR-28-5p antagomir were intravenously injected (5 nmol/mouse, twice a week) for 4 weeks. The cervical dislocation was used for mice euthanasia. All animal care and related experiments outlined in this study were performed in accordance with the Guidelines for the Care and Use of Laboratory Animals drafted by the US National Institutes of Health (NIH Publication, 8th Edition, 2011). Animal experiments were conducted following the Guidelines for Animal Experimentation of China Pharmaceutical University, and the protocols were approved by the Science and Technology Department of Jiangsu Province (SYXK (SU) 2016-0011).

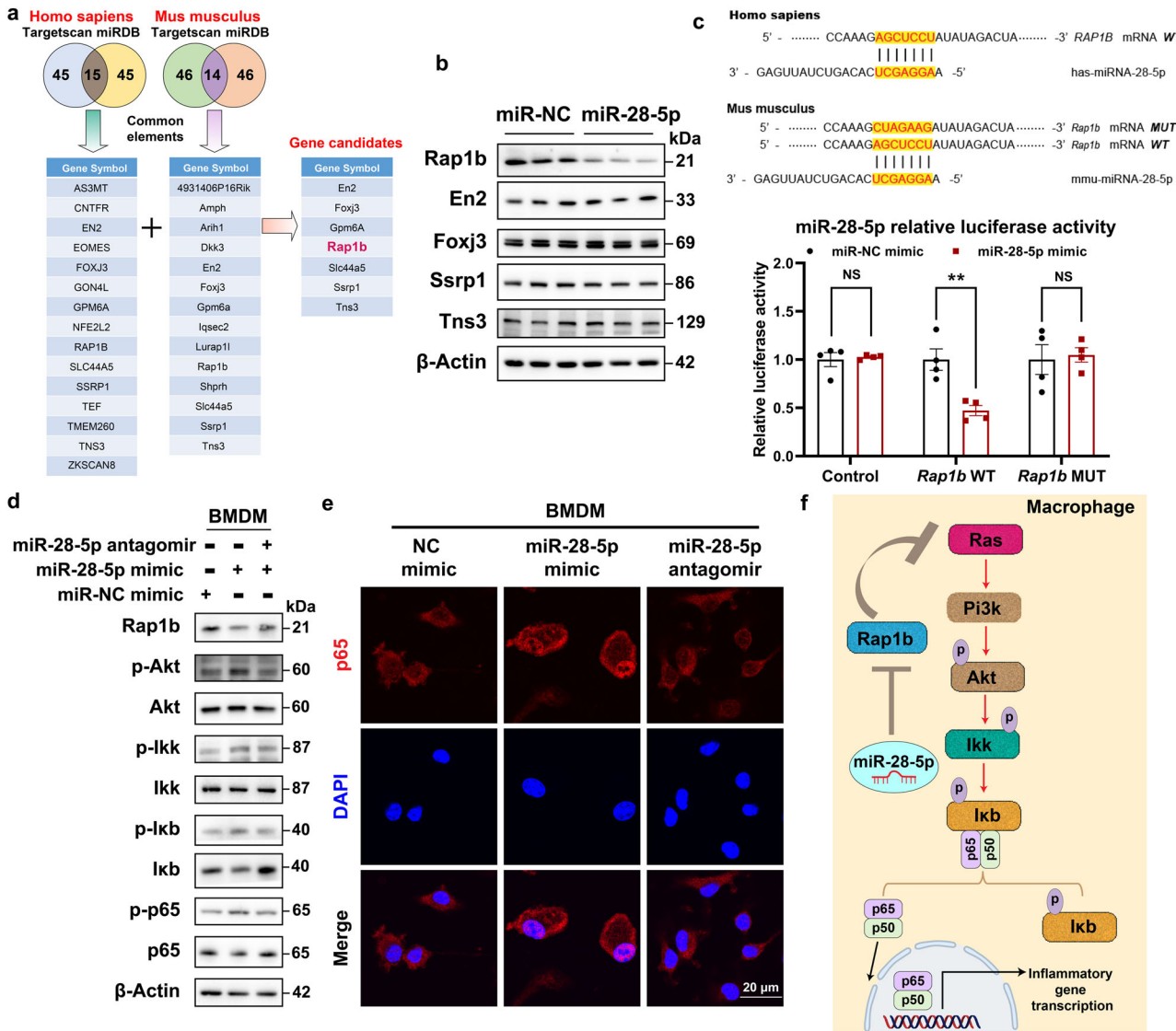

**Fig. 7 | miR-28-5p targets Rap1b to stimulate the expression of inflammatory cytokines via NF-κB. a** The potential targets of miR-28-5p were predicted by integrating the results of two databases (TargetScan and miRDB). **b** BMDMs were treated with miR-28-5p for 24 h, the putative targets were analyzed by western blot. **c** The predicted binding motif between miR-28-5p and RAP1B. The wild-type or a mutated binding site between miR-28-5p and Rap1b was cloned into pmirGlo vector. BMDMs were co-transfected with either miR-NC control or miR-28-5p mimics for 24 h, the luciferase activity was then measured (*n* = 4 independent experiments per group, miR-NC mimic versus miR-28-5p mimic, for control, *P* = 0.99, for *Rap1b* WT, *P* = 0.0020, for *Rap1b* MUT, *P* = 0.98). **d** BMDMs were treated with miR-28-5p mimics or miR-28 inhibitor for 24 h, proteins that affecting NF-κB signaling were analyzed by western blot. **e** BMDMs were treated with miR-28-5p mimics or miR-28 inhibitor for 24 h, cellular distribution of p65 were analyzed by immunofluorescence staining. **f** The signaling pathway through which miR-28-5p target Rap1b to stimulate the expression of inflammatory cytokines via NF-κB. Data are presented as mean ± SEM. *n* = 6 mice per group. *\*P* < 0.05, *\*\*P* < 0.01, *\*\*\*P* < 0.001, NS no significant difference, one-way ANOVA. Source data are provided in the Source Data file.

## Human specimens

Human blood specimens were collected at multiple centers by qualified medical staff. Nine individuals with NASH and 19 controls without NASH were enrolled without any age or sex preference. We excluded individuals with known blood-transmitted infectious diseases, chronic inflammatory systemic diseases, and other significant disease like severe cardiac, liver, or kidney diseases or tumors. Human liver samples from patients with NAFLD were diagnosed by abdominal ultrasound and verified for liver histology. The liver samples were from treatment-naïve patients following bariatric surgery to rule out the experimental complications due to medication. Normal human liver tissue comprised the uninvolved surrounding tissue and was obtained from NAFLD-free donors undergoing partial hepatectomy for hepatocarcinoma.

Information obtained from all participants and/or their relatives before sample collection was kept confidential. Informed consents were obtained from all participants and the experiments conformed to the principles outlined in the WMA Declaration of Helsinki and the Department of Health and Human Services Belmont Report. All participants received compensation. All procedures were approved by the Ethical Committee of Guangdong Provincial People's Hospital (Guangdong Academy of Medical Sciences) and the Beijing You'An Hospital.

## Cell lines

The Hepa 1–6, HepG2, and HEK293T cell lines were purchased from the Type Culture Collection of the Chinese Academy of Sciences (Shanghai, China). All the cell lines were examined for mycoplasma

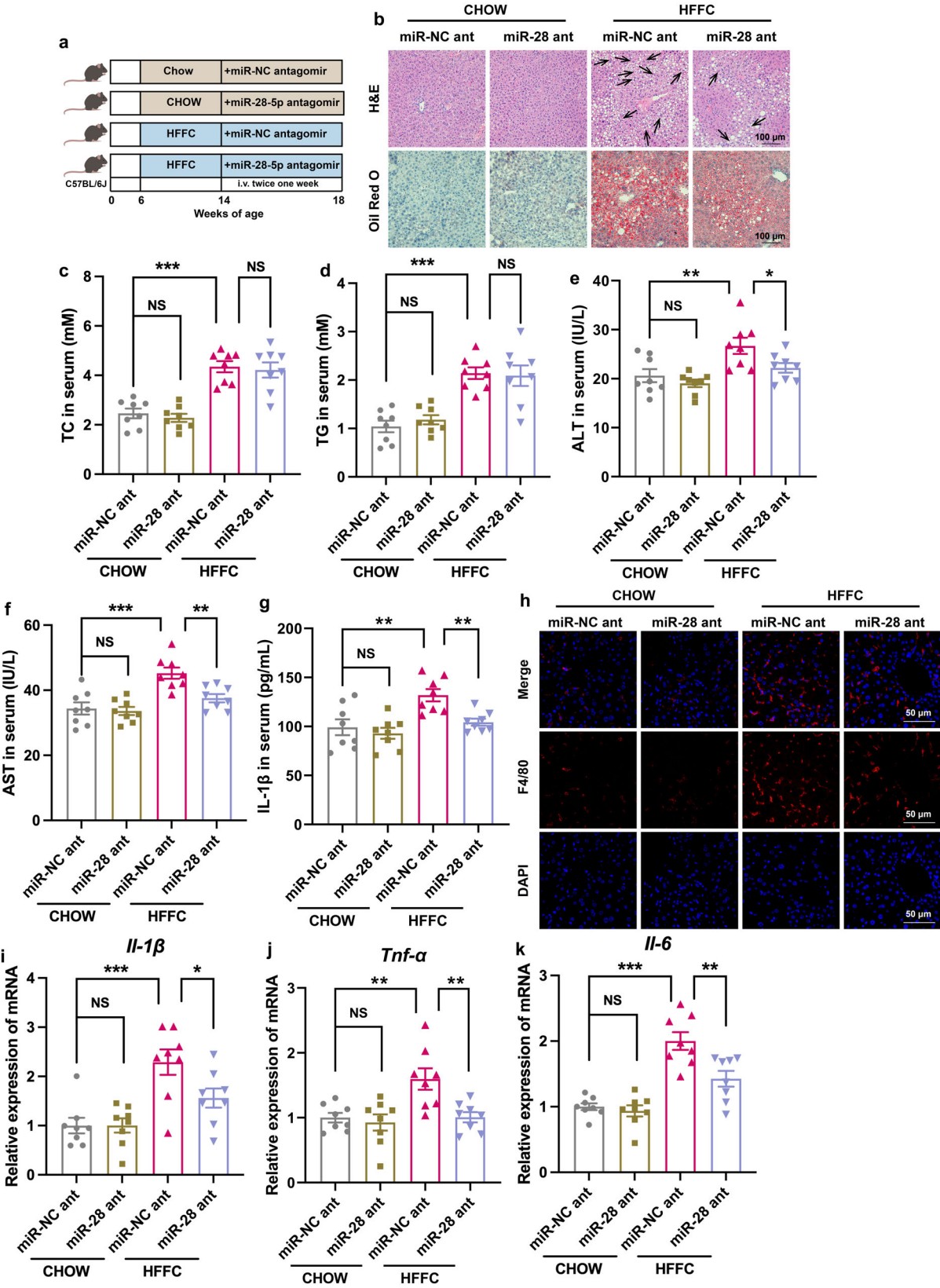

contamination, and the results were negative. The Hepa 1−6, HepG2, and HEK293T cells were cultured at 37 °C in a humidified atmosphere (5% $CO_2$ and 95% air) in Dulbecco's modified Eagle medium (DMEM, KeyGEN, China) supplemented with 10% fetal bovine serum (FBS, Gibco, USA), 100 units/mL penicillin, and 100 μg/mL streptomycin sulfate (KeyGEN, China).

**Isolation of primary hepatocytes**

Mice were perfused with buffer A (Calcium and magnesium-free Hanks containing 0.2 mM EGTA, 10 mM HEPES, 1 mM glucose and 0.2% BSA, pH 7.4) through the vena cava for 5 min. After the liver turned yellow-white in color, buffer B (DMEM with 1 mM magnesium, 1 mM calcium, 30 mM HEPES, and 0.5 mg/mL collagenase D)

**Fig. 8 | NPs carrying miR-28-5p antagomir attenuates inflammation in HFFC-diet-induced NASH mice. a** Experimental scheme of delivery of nanoparticles carrying miR-28-5p antagomir to macrophages in mice fed with HFFC diet. **b** Representative H&E stained FFPE liver sections and ORO stained frozen liver sections. Each arrow indicated hepatocyte ballooning. **c**–**g** Mice were fed with chow or HFFC diet for 8 weeks, nanoparticles carrying miR–NC or miR-28-5p antagomir were injected intravenously for 4 weeks. **c** Serum TC (chow+miR-NC ant versus chow+miR-28 ant, $P = 0.90$, chow+miR-NC ant versus HFFC+miR-28 ant, $P = 9.4E-06$, HFFC+miR-NC ant versus HFFC+miR-28 ant, $P = 0.96$), (**d**) serum TG (chow+miR-NC ant versus chow+miR-28 ant, $P = 0.83$, chow+miR-NC ant versus HFFC+miR-28 ant, $P = 2.7E-05$, HFFC+miR-NC ant versus HFFC+miR-28 ant, $P = 0.99$), (**e**) serum ALT (chow+miR-NC ant versus chow+miR-28 ant, $P = 0.70$, chow+miR-NC ant versus HFFC+miR-28 ant, $P = 0.0046$, HFFC+miR-NC ant versus HFFC+miR-28 ant, $P = 0.040$), (**f**) serum AST (chow+miR-NC ant versus chow+miR-28 ant, $P = 0.97$, chow+miR-NC ant versus HFFC+miR-28 ant, $P = 9.6E-05$, HFFC+miR-NC ant versus HFFC+miR-28 ant, $P = 0.0046$) and (**g**) serum IL-1β (chow+miR-NC ant versus chow+miR-28 ant, $P = 0.81$, chow+miR-NC ant versus HFFC+miR-28 ant, $P = 0.0019$, HFFC+miR-NC ant versus HFFC+miR-28 ant, $P = 0.0090$) of were analyzed after sacrifice at 18th week ($n = 8$ mice per group). **h** Representative liver sections stained with F4/80. The expression of (**i**) *Il-1β* (chow+miR-NC ant versus chow+miR-28 ant, $P = 0.99$, chow+miR-NC ant versus HFFC+miR-28 ant, $P = 1.9E-04$, HFFC+miR-NC ant versus HFFC+miR-28 ant, $P = 0.035$), (**j**) *Tnf-α* (chow+miR-NC ant versus chow+miR-28 ant, $P = 0.94$, chow+miR-NC ant versus HFFC+miR-28 ant, $P = 0.0032$, HFFC+miR-NC ant versus HFFC+miR-28 ant, $P = 0.0035$), (**k**) *Il-6* (chow+miR-NC ant versus chow+miR-28 ant, $P = 0.95$, chow+miR-NC ant versus HFFC+miR-28 ant, $P = 5.9E-07$, HFFC+miR-NC ant versus HFFC+miR-28 ant, $P = 0.0014$) genes in the liver of each group of mice was analyzed by qRT-PCR ($n = 8$ mice per group). Data are presented as mean ± SEM. $n = 6$ mice per group. *$P < 0.05$, **$P < 0.01$, ***$P < 0.001$, NS no significant difference, one-way ANOVA. Source data are provided in the Source Data file.

was used for perfusing the liver for another 5 min. Subsequently, perfusion was terminated and the liver was excised using ice-cold DMEM. Cells from digested livers were filtered through a 100 μm cell strainer, and centrifuged at 60 g for 6 min at 4 °C. After washing twice with DMEM, Percoll premixed with 10×PBS was added to a final concentration of 50% and the sample was centrifuged at 100 g for 10 min at 4 °C. After removing the supernatant, the hepatocyte pellet was washed once with DMEM and cultured in DMEM supplemented with 10% FBS.

### Isolation of Kupffer cells
Mice were perfused and digested as method "isolation of primary hepatocytes". As described[79], cells from digested livers were filtered through a 100 μm cell strainer, and centrifuged at 60 g for 6 min at 4 °C. The supernatant was collected for use. In a 50 mL Falcon tube, slowly add 20 mL of 25% Percoll on top of 14.5 mL of 50% Percoll. Then the supernatant was layered on the Percoll gradient with and centrifuged at 1200 g for 30 min at 4 °C without brakes. The middle interphase (white cell ring) was carefully collected and transferred into a new 50 mL tube. The cells were washed twice with PBS and centrifuged at 50 g for 10 min at 4 °C. Cells were resuspended in DMEM supplemented with 10% FBS and cultured for 30 min, followed by washing the cells with PBS and adding fresh medium to remove the nonadherent cells.

### Isolation of hepatic stellate cells (HSC)
Mice were perfused and digested as method "isolation of primary hepatocytes". As described[80], cells from digested livers were filtered through a 100 μm cell strainer, and centrifuged at 580 g for 10 min at 4 °C. The supernatant was aspirated until 10 mL remaining in the tube, filled up to 32 mL with GBSS/B and then added 16 mL of Nycodenz solution. Mixed thoroughly by gently inverting the tube and pipetted 12 mL of cell-Nycodenz suspension into each of the four 15 mL Falcon tubes. Gently overlayed the cell-Nycodenz suspension with 1.5 mL of GBSS/B and centrifuged at 1380 g for 17 min at 4 °C without brake. After the centrifugation, the HSCs was collected which were visible as a thin white layer in the interface between the cell-Nycodenz solution and the overlay with GBSS/B. Added GBSS/B to fill up the tube to 50 mL and gently pipetted to resuspend the harvested HSCs, centrifuged at 580 g for 10 min at 4 °C. Aspirated the supernatant and cultured cell pellet in DMEM with 10% FBS.

### Isolation of bone marrow-derived macrophages (BMDMs)
Bone marrow suspensions from each set of long bones (1 tibia and 1 femur) were passed through a 100 μm cell strainer. BMDMs were seeded at a density of $1 \times 10^6$ cells per well in a six-well plate or in glass cell dishes (NEST, Wuxi, China). Cells were cultured for 7 days at 37 °C in a humidified atmosphere with 5% $CO_2$ in α-MEM containing 10% FBS, and 30 ng/mL Macrophage colony-stimulating factor 1 (M-CSF).

### Gene knockdown and overexpression
To demonstrate the impacts of SHH pathway on HSP90β protein levels, primary hepatocytes, Hepa 1–6 or HepG2 cells were transfected with indicated siRNA (Genepharma, Shanghai, China) using RNAiMAX (Invitrogen, California, USA) following the manufacturer's instructions. To perform co-immunoprecipitation or in vitro deubiquitination assay, indicated plasmids were transfected with Lipofectamine 3000 in primary hepatocytes, Hepa 1–6 or 293 T cells following the manufacturer's instructions. To study the role of miR-28-5p in regulating inflammation in macrophages, BMDM cells were transfected with miR-28-5p mimic or miR-28-5p antagomir using RNAiMAX (Invitrogen, California, USA) following the manufacturer's instructions.

### Hepatic organoid culture
As described previously[81], isolated liver cells of *Hsp90β^{fl/fl}* or *Hsp90β^{ΔHep}* mice were counted and mixed with Matrigel in suspension plates (Greiner). 50,000 cells were used per well of a 24 well plate. After Matrigel was solidified, culture Medium was added. Culture Medium consists of AdDMEM/F12 (Thermo Scientific, with HEPES, GlutaMax and Penicillin-Streptomycin) plus 15% RSPO1 conditioned medium, B27 (minus vitamin A), 50 ng/ml EGF (Peprotech), 1.25 mM N-acetylcysteine (Sigma), 10 nM gastrin (Sigma), 3 μm CHIR99021 (Sigma), 25 ng/ml HGF (Peprotech), 50 ng/ml FGF7 (Peprotech), 50 ng/ml FGF10 (Peprotech), 1 μM A83-01 (Tocris), 10 mM Nicotinamide (Sigma), and 10 μM Rho Inhibitor γ–27632 (Calbiochem). 7 days after seeding, organoids were cultured with PA, OA or Shh. Two days later, miR-28-5p antagomir was added for another 2 days. Organoids were collected and inflammatory genes expression was detected by qRT-PCR.

### Western blot analysis
Cells or tissues were homogenized in RIPA buffer (65 mM Tris-HCl pH 7.5, 150 mM NaCl, 1 mM EDTA, 1% NP-40, 0.5% sodium deoxycholate and 0.1% SDS) supplemented with protease (Roche, 04693116001) and phosphatase inhibitors (Roche, 04906837001). Equal amounts of cell lysates containing proteins (30–40 mg of protein per lane) from each biological replicate were analyzed by western blotting. Using the Tanon 5200 imaging system (Tanon, China), the protein bands on the blots were detected with Femto-sig ECL western blotting substrate (Tanon). Western blot data in figures and supplemental figures are all representatives of at least three independent experiments.

### Co-immunoprecipitation (Co-IP) assay
Whole-cell extracts were prepared in IP lysis buffer (20 mM Tris-HCl, pH 7.5, 137 mM NaCl, 5 mM EDTA, 1% NP-40, 10% glycerol, 50 mM NaF, 1 mM $Na_3VO_4$, and protease inhibitor PMSF) containing protease and phosphatase inhibitors. Approximately 10% of the supernatant was used for western blotting as the input, while the rest of the homogenates were incubated with indicated antibodies (Myc or IgG)

overnight at 4 °C. Protein A/G plus agarose beads (ThermoFisher, #20422) were added and incubated at 4 °C for another 2 h. The beads were washed with cold PBS five times, followed by western blotting.

## In vitro deubiquitination assay

As described in previous report[39], 293 T cells were transfected with plasmid Myc-HSP90β and HA-ubiquitin for 48 h. After that, cells were lysed and immunoprecipitated with anti-Myc antibody for use. Hepatocyte cytosols were prepared from Hepa 1–6 cells or primary hepatocytes after the treatment of vehicle or Shh, or from primary hepatocytes of mice fed with chow/HFFC diet for 16 weeks. The pellet ubiquitin samples from 293 T cells were then incubated with hepatocyte cytosols above for 1 h min at 37 °C, and the reaction was stopped by adding loading buffer (Yeasen, #20315ES05), followed by SDS-PAGE and analysis by immunoblotting.

## Quantitative RT-PCR analysis

Total RNA was extracted following the RNA extraction protocol according to the manufacturer's instructions (Vazyme, China). cDNA was synthesized using SuperScript III and random hexamers. AceQ qPCR SYBR Green Master Mix (Vazyme, #Q111-02/03) was used to quantify the PCR-amplified products. The levels of mRNA expression of the target genes were normalized to that of *Gapdh*.

For miRNA RT-PCR analysis, cDNA was synthesized using TaqMan microRNA reverse transcription kit and miRNA primers (Genepharma, China). qPCR was performed using TaqMan universal master mix II and miRNA primers (Genepharma, China). qRT-PCR data were normalized using a robust global median normalization method described previously[81].

## Chromatin-immunoprecipitation (ChIP) assay

ChIP samples from Hepa 1–6 cells treated with vehicle or Shh were prepared following instructions specified in the SimpleChIP® Plus Sonication Chromatin IP Kit (CST, #56383). Nuclear samples were incubated with the BMAL1 (CST, 14020), SREBP1 (Abcam, ab28481), or IgG (CST, 2729) antibody overnight.

## Electrophoretic mobility shift assay (EMSA)

The nuclear extracts from primary hepatocytes were prepared for DNA-protein binding reactions. The biotin-labeled wild-type and mutated oligonucleotides for EMSA probes were procured from GENEWIZ (Suzhou, China) (Supplementary Table 1). DNA-protein binding reactions were conducted using an EMSA/Gel-Shift Kit (Beyotime) following the manufacturer's instructions. Samples were loaded onto 10% ExpressPlus™ PAGE Gel (GenScript) after the addition of 5 μL loading buffer. After electrophoresis at 100 V in 0.5× TBE buffer (0.045 mol/L Tris-boric acid, 0.001 mol/L EDTA) for 45 min, an electrotransfer was performed on a positively charged nylon membrane (Beyotime) in 0.5× TBE buffer at 380 mA for 45 min. The membranes were cross-linked with a UV linker at 254 nm and 120 mJ/cm² for 2 min. The bands in immunoblots were visualized using the Chemiluminescent EMSA Kit (Beyotime).

## Exosomal isolation and characterization

Liver tissues from *Hsp90β^fl/fl* or *Hsp90β^ΔHep* mice were obtained and cut into small pieces. Liver tissues were cultured in serum-free DMEM (5 mL DMEM, 0.5 mg/mL collagenase D) for 90 min and maintained in an incubator at 37 °C with 5% $CO_2$/95% $O_2$ humidified atmosphere. Exosomes were purified from the supernatant by the differential centrifugation method described previously[15]. Briefly, supernatants of the cultured liver tissues were collected and centrifuged at 3000 g for 15 min to remove any tissue debris, followed by centrifugation at 10,000 g for 30 min for the removal of large vesicles. Next, the supernatants were centrifuged at 110,000 g for 90 min. The exosomal pellet was collected and resuspended in PBS. The protein

concentrations in exosomes were measured using a BCA protein assay kit. All steps were performed at 4 °C.

Exosomes were isolated from cell culture or human plasma via differential centrifugation as described above. For their characterization, particle size analysis was performed on the NanoSight instrument (Malvern Instruments). The morphology of exosomes was observed using a transmission electron microscope (TEM) (HT7700, Hitachi), and the protein content was determined by western blotting.

## miRNA sequencing and analysis

In total, 5 ng RNA from exosomes per sample was used for small RNA library preparation. Sequencing libraries were generated using NEBNext® Multiplex Small RNA Library Prep Set for Illumina® (NEB, USA). The library quality was assessed on an Agilent Bioanalyzer 2100 system using DNA High Sensitivity Chips and sequenced on an Illumina HiSeq 2500/2000 platform. Known miRNAs were identified by comparison with the sequences in the miRBase 20.0 database (https://mirbase.org/)[82]. Differential expression analysis (exosomes from chow-diet group vs. exosomes from HFFC-diet group) was performed using the DEGseq (2010) package in R. Small RNA sequencing and analysis were performed by NovogeneCo., Ltd. (Beijing, China).

## miR-28-5p target gene prediction and validation

TargetScan and miRDB were used to predict the target genes of miR-28-5p. To validate the binding between miR-28-5p and *Rap1b* mRNA, wild-type or mutated sequences of the predicted binding site were cloned into pmirGLO plasmid. BMDMs were transfected with a luciferase reporter construct along with a miR-28-5p mimic or empty vector. After transfection for 24 h, the luciferase activities were detected using a dual luciferase kit (Promega, USA) following the manufacturer's instructions.

## RNA-seq library preparation and sequencing

Oligo(dT)[6]-attached magnetic beads were used to purify mRNA. Purified mRNA was fragmented into small pieces using a fragment buffer at an appropriate temperature. Subsequently, the first-strand cDNA was generated using random hexamer-primed reverse transcription, followed by a second-strand cDNA synthesis. Afterward, A-Tailing Mix and RNA Index Adapters were added for end repair by incubation. The cDNA fragments obtained from the previous step were amplified by PCR, and products were purified using Ampure XP Beads, and dissolved in an ethidium bromide solution. The product was validated for quality control on the Agilent Technologies 2100 bioanalyzer. The double-stranded PCR products were heat-denatured and circularized using the splint oligo sequence to obtain the final library. The single-strand circle DNA (ssCir DNA) was the final library. The final library was amplified with phi29 to prepare a DNA nanoball (DNB) comprising more than 300 copies of one molecule. DNBs were loaded into the patterned nanoarray and single-end 50 base reads were generated on the BGIseq500 platform (BGI-Shenzhen, China).

## Immunofluorescence staining

Liver tissues were snap-frozen at the optimum cutting temperature on dry ice. Six μm cryo-sections of tissues were cut and fixed in pre-cold acetone for 20 min. Slides were blocked with 5% normal donkey serum for 60 min at RT. After washing, nuclei were stained with DAPI for 10 min at RT. Mounting media and coverslips were then placed onto slides for imaging. Images were captured under a fluorescence microscope (Nikon, Japan). Cells were fixed with acetone for 10 min, followed by blocking with 1% BSA and staining with 0.5 mg/mL DAPI for 10 min at RT. The coverslips were mounted and cells were evaluated by fluorescence microscopy. Images were captured using a fluorescence microscope (Nikon, Japan).

## Nanoparticle (NP) preparation and characterization

NP was prepared as described previously[83,84]. miR-28-5p antagomir with sm-HA was mixed with protamine (N:P = 40:1, mol: mol; protamine: sm-HA = 1: 3.75, w- w), and incubated for 10 min. N,N'-bis (acryloyl) cystamine (BAC) dissolved in dimethyl sulfoxide (DMSO) was then added to the mixture (BAC: protamine = 100: 1, mol: mol). Polymerization was initiated by adding GSH (10 mM), ammonium persulfate, and tetramethylethylenediamine. After 1 h, the solution was washed thrice using diethyl pyrocarbonate (DEPC) water, and NPs were obtained. The particle size of NP was characterized on a zeta sizer (NanoZS90, Malvern). The miRNA-packing capacity was evaluated using the agarose gel electrophoretic assay. Mice were intravenously injected with NPs containing Cy5- miR-28-5p antagomir. The mice were euthanatized 24 h after the injection, and the major tissues were excised and imaged using IVIS (IVIS Spectrum, PerkinElmer) at predetermined time points.

## Statistics and reproducibility

All quantitative measurements (i.e., weights, serum assays, qRT-PCR) have at least three independent biological repeats. The results were expressed with mean and standard error of the mean (SEM) unless mentioned otherwise. All statistical analyses were performed using GraphPad Prism software (version 8.0.2). Simple two-tailed Student's t-tests were used for experimental setups requiring testing of just two conditions. For comparisons between more than two conditions, one-way ANOVA with Dunnett's correction (when several conditions were compared to one condition) or Tukey's correction (when several conditions were compared to each other) were used. The criterion for statistical significance was set at $P < 0.05$. We collected data from the animal studies in a blinded manner, and no data were excluded from the final statistical analysis.

## Reporting summary

Further information on research design is available in the Nature Portfolio Reporting Summary linked to this article.

## Data availability

The raw RNA-seq data are available for download from NCBI Gene Expression Omnibus under the accession code GSE232758 and GSE232759. The data that support the findings of this work are available from the corresponding author on request. There are no restrictions on data availability in the current work. Source data are provided with this paper.

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

## Acknowledgements

We thank Prof. Yun Zhao at Center for Excellence in Molecular Cell Science, University of Chinese Academy of Sciences, Chinese Academy of Sciences for providing us Shh plasmids and helpful discussions. We thank Prof. Li Qiang at Columbia University for critical reading of our manuscript. This work was supported by National Natural Science Foundation of China (82273990 to X.X.) and the National Key R&D Program of China (Grant no. 2023YFA1801100 to X.X.), National Key Research and Development Program of China (2022YFF0710600 to Z.Y.), CAMS Innovation Fund for Medical Sciences (2016-I2M-4-001 to PP.L.), Beijing Outstanding Young Scientist Program (BJJWZYJH01201910023028 to PP.L.), the Chinese Academy of Medical Sciences (CAMS) Central Public-interest Scientific Institution Basal Research Fund (2018RC350004 and 2017PT31046 to PP.L.).

## Author contributions

X.X. conceived the project. X.X. and Z.Y. designed the experiments. W.Z., J.L., L.F., H.X., S.S. performed the experiments. J.L., S.L., J.K., Z.Y. provided experimental and clinical resources. W.Z., J.L., P.L., PP.L., Z.Y. and X.X. analyzed the data. W.Z. and X.X. wrote the paper with input from all authors.

## Competing interests

X.X., W.Z. and L.F. are applying for two patents related to this work. The rest of the authors have no financial or non-financial competing interests.
