## [Peer Review File · Nature Communications]

REVIEWER COMMENTS

Reviewer #1 (Remarks to the Author):

This is a very interesting and novel study revealing that SHH promotes NASH process through HSP90 β . To demonstrate this point, the authors used different mouse models, as well as patient samples to find out specific role of HSP90 β in NASH development. The authors provided extensive set of data showing that USP31, which is transcriptionally regulated by Shh-Bmal1, was a novel DUB of HSP90 β . The authors also found HSP90 β increased secretion of exosomes enriched with miR-28-5p, which promoted inflammatory response in macrophages. Clinical data also showed serum miR-28-5p correlated quite well with NASH. Finally, the authors used nanoparticles wrapping miR-28-5p antagomir decrease inflammation in HFFC diet-induced NASH mouse. Overall, I find the study to be compelling and well suited for Nature Communications. However, there are a few points that should be addressed prior to publication.

Major points:

1. Since activation of the Shh pathway promoted Bmal1's nuclear translocation, is there any relevance between circadian rhythm and NASH? Please discuss this.
2. Homemade nanoparticles in this manuscript is characterized with good miRNA-packing capacity and particle size, mainly accumulating in the liver. The authors need to briefly discuss the ability of NPs to enter macrophages in the liver.
3. Is there any researches about whether miR-28-5p has effects on hepatocytes or hepatic stellate cells. If there any, do these known miR-28-5p targeted genes participate in NASH process?
4. To detect miR-28-5p in human samples, exosomes are extracted from serum at the first. This step may increase the instability of detected data. Can miR-28-5p be extracted and detected directly from serum without affecting the accuracy of the data?

Minor points:

Page 6 lines 162, "absorbed excess lipids", "excess" can be removed.

Page 6 lines 168, "caused" should be "causing"

Page 7 lines 175, "in mouse blood", do the authors mean "in the serum"?

Page 7 lines 179 and Page 8 lines 215, "Compared to" should be "Compared with"

Page 7 lines 201, "I Importantly" should be "Importantly"

Page 13 lines 355, Abbreviations appearing for the first time (NPs) should be given in full text (nanoparticles)

Page 13 lines 356, “The particle size of NPs was characterized were approximately 100 nm” needs to be revised.

Reviewer #2 (Remarks to the Author):

Comments to Authors

This manuscript investigates the role of HSP90 β in Sonic hedgehog (SHH) mediated NASH development. The authors extend their previous studies on investigating SHH signaling mechanisms to identify downstream mediators. Here the investigators show that HSP90 β is critical in SHH mediated fatty liver development and inflammation. The authors utilize in vivo and in vitro strategies to unravel the role of SHH-HSP90 β -miR-28-5p axis in NASH, however several concerns listed below, in methodologies and lack of clarity in results presented, dampen enthusiasm. Overall the data presented lack robust outcomes to justify hepatocyte specific SHH-HSP90 β signaling in NASH.

1) Experiments in Figure 1 utilize cyclopamine, pharmacological inhibitor of SMO, to determine the effect of SHH signaling on murine NASH development. Using this inhibitor, investigators also evaluate alterations in HSP90 β in Figure 2. Additional specific inhibition strategies of SHH signaling such as siRNA must be used for robust analysis, since identification of SHH-specific HSP90 β changes is based on this inhibition.

2) In Figure 2, in vivo and in vitro approaches show that HSP90 β protein is increased by SHH in HFFC-livers and primary hepatocytes as well as HEPA 1-6 cells and HEPG2 cells. However, it is not clear if this increase in HSP90 β occurs in hepatocytes only. Whether HSP90 β is altered in liver macrophages and stellate cells must be evaluated. It is important to demonstrate cell-type specificity in HSP90 β to justify subsequent generation of hepatocyte-specific HSP90 β KO mice.

3) Figure 2 I-N shows HSP90 β and HSP90 α levels in human liver samples. It is not clear if the liver samples are NAFLD or NASH patients? Also lack of representative IHC micrographs make it difficult to discern the levels shown in the graphs.

4) HSP90 β flox/flox mice were generated commercially. Data must be shown to confirm specificity of these mice. Further to genotype hepatocyte-specific HSP90 β KO mice Fig S2B confirms that HSP90 β is not expressed in liver tissues of KO mice. This is intriguing since HSP90 β is a constitutive form of HSP90 and expressed in several cell types in the liver including macrophages. To confirm hepatocyte specific KO, HSP90 β must be decreased or lost in hepatocytes whereas other liver cells should exhibit HSP90 β expression.

- 5) Figure S5 utilizes 17-AAG in vivo as a HSP90 inhibitor to confirm the specificity of HSP90 β in murine NASH. 17-AAG can inhibit HSP90 α and HSP90 β in various cell types. Additional strategies to target HSP90 β using siRNA must be employed to determine SHH signaling in hepatocytes.
- 6) Overexpression of SHH using ADV-SHH was demonstrated by detection of GFP+ signals in livers (Fig S6A). It is not clear which cells overexpress SHH and the magnitude of overexpression?
- 7) Much of the work on HSP90 β deubiquitylation is performed in hepatocyte cell lines. These studies must be confirmed in primary hepatocytes and changes in HSP90 β deubiquitylation must be demonstrated in NASH hepatocytes. EMSA analysis using BMAL1 antibodies (Fig 5D) is not convincing.
- 8) SHH induced inflammation in NASH is attributed to HSP90 β facilitated extracellular communication by liver exosomes. The protocol used to prepare liver exosomes is not clear. This group has previously published preparation of exosomes from adipose tissue and the same method is extended to the liver. It is not clear if these purified liver exosomes capture the disease related phenotype.

Reviewer #3 (Remarks to the Author):

Your elegant studies reveal a novel Hedgehog signaling mechanism that has potentially broad implications, including induction of a pro-inflammatory macrophage phenotype that contributes to the pathogenesis of nonalcoholic steatohepatitis (NASH) and related liver fibrosis.

Briefly, your work complements and extends earlier work by other groups showing that both hepatic expression of SHH ligand and SHH-initiated down-stream signaling that activates Smoothed (Smo) are increased in people and mice with NASH. Importantly, your studies show that systemic administration of cyclopamine (Cp), a direct inhibitor of Smo, improves diet-induced steatosis, inflammation and fibrosis in mouse models. Although not cited in your manuscript, the inhibitory effects of direct Smo antagonism on murine liver inflammation and fibrosis had been reported previously and attributed to paracrine effects of hepatocyte-derived HH ligands on other types of HH-responsive liver cells (PMID: 21912653). However, the earlier study did not investigate the possible role of autocrine hepatocyte HH signaling in NASH pathogenesis.

Your work demonstrates that SHH-initiated activation of Smo in hepatocytes permits Bmal1 (a circadian clock-associated transcription factor) to disassociate from SuFu. Your result complements and extends data from another group which showed that hepatocyte Smo activity regulates the circadian clock

(PMCID PMC6146234). You have significantly advanced understanding of the mechanisms involved by demonstrating that following Smo activation and release from SuFu, "free" Bmal1 localizes to nuclei where it promotes the transcription of a deubiquitinase (Usp 31) that stabilizes the heatshock chaperone, Hsp90b. Hsp90b, in turn, facilitates the release of hepatocyte-derived exosomes carrying miR 28-5p and this miR suppresses Rap1b in macrophages, resulting in activation of NF-KB signaling and increased macrophage production of several pro-inflammatory cytokines that are known to promote insulin resistance, hepatic steatosis, lipotoxicity and fibrogenic activation of hepatic stellate cells. Based on these results and evidence that SHH and miR28-5p are increased in humans with NASH, you suggest that miR28-5p might be both a novel biomarker and therapeutic target in NASH. The data are compelling and as you state, have actionable implications clinically. However, a few issues require clarification:

1 - Fig 2I is confusing as presented. More granular information is necessary to assess correlations between expression of SHH and Hsp90B and NASH severity. For example, how do levels of SHH/Hsp90b correlate with the severity of the NAFLD Activity Score (NAS) and fibrosis stage? The NAS is typically assessed to judging the severity of hepatic steatosis (graded 0-3), liver inflammation (graded 0-2) and hepatocyte ballooning (graded 0-2). Fibrosis staging (generally scored 0-4) is used to assess fibrosis severity. This information is particularly important for your studies because SHH is known to accumulate in ballooned hepatocytes and the severity of hepatocyte ballooning and liver fibrosis severity are significantly correlated.

2 - You suggest that circulating miR28-5p might be a novel biomarker for NASH. You also show that serum levels of SHH increase in NASH and that SHH initiates the process that leads to increased miR28-5p. Is miR28-5p superior to SHH for "predicting" severity of NASH or fibrosis?

3 - Some of the approaches that you used to manipulate HH signaling (e.g., systemic administration of Cp or Adenoviral vector delivery of SHH ligand) are not hepatocyte-specific and so, would impact pathway activity in multiple cell types both in liver and other tissues. While your in vitro experiments clearly demonstrate that SHH can induce Hsp90b in hepatocytes and your in vivo experiments prove that hepatocyte derived Hsp90b critically mediates liver inflammation and fibrosis, can you assure that SHH is the main mediator of Hsp90b accumulation in hepatocytes in your mouse NASH models (or human NASH liver samples)? What was the rationale for using ADV-SHH in mice with diet-induced NASH given that your data and work of others shows that endogenous SHH expression is increased in NASH?

4 - Previous publications have shown that liver cell-derived exosomes carry SHH and Indian Hedgehog (PMCID PMC3724240). Studies have also reported that hepatocyte-derived IHH is important for activating macrophages and promoting hepatic stellate cell activation in mouse models of NASH (PMCID PMC5226184). Do the exosomes that carry miR25p also carry either of the HH ligands? If so, is the HH ligand cargo required to miR28-5p to promote the proinflammatory macrophage phenotype?

4 - Can you rectify your findings which suggest that stimulating Smo activity in hepatocytes promotes hepatic lipid accumulation with recent work from at least two other groups demonstrating that targeted inhibition of hepatocyte Smo promotes hepatocyte lipid accumulation (PMCID PMC4869931, PMID PMC8559255)?

5 - Others have reported that Smo activity inhibits adipogenesis but you found that systemic exposure to Cp (a Smo antagonist) reduced adiposity in mice with diet-induced obesity and NASH. How might this paradox be explained?

We are grateful for the reviewers' constructive comments and suggestions and particularly appreciated that a number strengths were identified in our work. Here, we have attempted to address all the reviewers' concerns from the original submission point-to point. The reviewers' comments are provided in *bold italics* below with our response following.

Reviewer #1:

This is a very interesting and novel study revealing that SHH promotes NASH process through HSP90/3. To demonstrate this point, the authors used different mouse models, as well as patient samples to find out specific role of HSP90/3 in NASH development. The authors provided extensive set of data showing that USP31, which is transcriptionally regulated by Shh-Bmal1, was a novel DUB of HSP90/3. The authors also found HSP90/3 increased secretion of exosomes enriched with miR-28-5p, which promoted inflammatory response in macrophages. Clinical data also showed serum miR-28-5p correlated quite well with NASH. Finally, the authors used nanoparticles wrapping miR-28-5p antagomir decrease inflammation in HFFC diet-induced NASH mouse. Overall, I find the study to be compelling and well suited for Nature Communications. However, there are a few points that should be addressed prior to publication. Major points:

We appreciate the reviewer's deeply insightful evaluation. We explained the all the questions reviewer concerned and some of them were added in discussion of our manuscript.

1. Since activation of the Shh pathway promoted Bmal1's nuclear translocation, is there any relevance between circadian rhythm and NASH? Please discuss this.

Circadian misalignment has been identified as a risk factor for metabolic

disease (Reinke and Asher, 2016). The circadian clock is involved in regulation of hepatic triglyceride accumulation, inflammation, oxidative stress, and mitochondrial dysfunction (Adamovich et al., 2014; Hatori et al., 2012; Jacobi et al., 2015). Bmal1, as a core circadian clock gene, play an important role in NAFLD. Bmal1 deficiency protects against steatohepatitis, inflammation, and fibrosis, preventing the development of NASH (Jouffe et al., 2022). We added some discussion in the revised manuscript.

2. Homemade nanoparticles in this manuscript is characterized with good miRNA-packing capacity and particle size, mainly accumulating in the liver. The authors need to briefly discuss the ability of NPs to enter macrophages in the liver.

Some studies suggested that nanoparticles can undergo transcytosis through cells in the liver, especially in Kupffer cells (Soji et al., 1992; Ogawara et al., 1999). The method of nanoparticle production in this manuscript has been described in our previous work. Our previous data demonstrated that the nanoparticle boosted the accumulation of its cargo in the liver of the HFD-fed obese mice, particularly in the activated macrophages (Shen et al., 2020).

3. Is there any researches about whether miR-28-5p has effects on hepatocytes or hepatic stellate cells. If there any, do these known miR-28-5p targeted genes participate in NASH process?

It was reported that miR-28-5p was downregulated in tumors (Lv et al., 2019) and cancer stem cells (Xia et al., 2019). IGF1 has been found as the direct target of miR-28-5p (Xia et al., 2019; Shi and Teng, 2015). The level of IGF-1 is down-regulated in NAFLD patients compared to healthy controls (Yao et al., 2019), this is in consistent with our finding that miR-28-5p was upregulated in NAFLD/NASH patients (Fig. 6h). In addition, miR-28-5p upregulated the expression of ATP-binding cassette transporter A1 (ABCA1)

via the inhibition of ERK2 in HepG2 cells (Liu et al., 2016). The plasma levels of miR-28-5p were significantly increased in unstable angina patients (Liu et al., 2015). However, the correlation between miR-28-5p and NASH has not been reported.

4. To detect miR-28-5p in human samples, exosomes are extracted from serum at the first. This step may increase the instability of detected data. Can miR-28-5p be extracted and detected directly from serum without affecting the accuracy of the data?

We used miRNeasy Serum/Plasma Advanced Kit (Qiagen, 217204) to extract total serum miRNAs. In the serum, there are various extracellular vesicles, including exosomes, microvesicles, microparticles, apoptotic bodies. In this manuscript, we found that miR-28-5p in hepatocytes-derived exosomes promoted inflammatory gene expression in macrophages. HSP90 is critical for the maturation and secretion of exosomes, while no studies have shown that HSP90 affects the secretion of other extracellular vesicles. To avoid the contamination of miRNAs from other extracellular vesicles, we extracted miRNAs directly from purified exosomes.

Minor points:

Page 6 lines 162, “absorbed excess lipids”, “excess” can be removed.

Page 6 lines 168, “caused” should be “causing”

Page 7 lines 175, “in mouse blood”, do the authors mean “in the serum”?

Page 7 lines 179 and Page 8 lines 215, “Compared to” should be “Compared with”

Page 7 lines 201, “I Importantly” should be “Importantly”

Page 13 lines 355, Abbreviations appearing for the first time (NPs) should be given in full text (nanoparticles)

Page 13 lines 356, “The particle size of NPs was characterized were approximately 100 nm” needs to be revised.

We are very grateful to the reviewer for reviewing the manuscript carefully and discovering some of the mistakes. We revised the text accordingly in the new manuscript.

Reference

- Reinke H, Asher G. Circadian Clock Control of Liver Metabolic Functions. *Gastroenterology* 150, 574-580 (2016).
- Adamovich Y, et al. Circadian clocks and feeding time regulate the oscillations and levels of hepatic triglycerides. *Cell Metab* 19, 319-330 (2014).
- Hatori M, et al. Time-restricted feeding without reducing caloric intake prevents metabolic diseases in mice fed a high-fat diet. *Cell Metab* 15, 848-860 (2012).
- Jacobi D, et al. Hepatic Bmal1 Regulates Rhythmic Mitochondrial Dynamics and Promotes Metabolic Fitness. *Cell Metab* 22, 709-720 (2015).
- Jouffe C, et al. Disruption of the circadian clock component BMAL1 elicits an endocrine adaption impacting on insulin sensitivity and liver disease. *Proc Natl Acad Sci U S A* 119, e2200083119 (2022).
- Soji T, Murata Y, Ohira A, Nishizono H, Tanaka M, Herbert DC. Evidence that hepatocytes can phagocytize exogenous substances. *Anat Rec* 233, 543-546 (1992).
- Ogawara K, et al. Uptake by hepatocytes and biliary excretion of intravenously administered polystyrene microspheres in rats. *J Drug Target* 7, 213-221 (1999).
- Shen S, et al. Collaborative assembly-mediated siRNA delivery for relieving inflammation-induced insulin resistance. *Nano Research* 13, 2958-2966 (2020).
- Lv Y, Yang H, Ma X, Wu G. Strand-specific miR-28-3p and miR-28-5p have differential effects on nasopharyngeal cancer cells proliferation, apoptosis, migration and invasion. *Cancer Cell Int* 19, 187 (2019).
- Xia Q, et al. MicroRNA-28-5p Regulates Liver Cancer Stem Cell Expansion via IGF-1 Pathway. *Stem Cells Int* 2019, 8734362 (2019).
- Shi X, Teng F. Down-regulated miR-28-5p in human hepatocellular carcinoma correlated with tumor proliferation and migration by targeting insulin-like growth factor-1 (IGF-1). *Mol Cell Biochem* 408, 283-293 (2015).
- Yao Y, et al. Insulin-like growth factor-1 and non-alcoholic fatty liver disease: a systemic review and meta-analysis. *Endocrine* 65, 227-237 (2019).
- Liu J, et al. MicroRNA 28-5p regulates ATP-binding cassette transporter A1 via inhibiting extracellular signal-regulated kinase 2. *Mol Med Rep* 13, 433-440 (2016).
- Liu J, et al. miR-28-5p Involved in LXR-ABCA1 Pathway is Increased in the Plasma of Unstable Angina Patients. *Heart Lung Circ* 24, 724-730 (2015).

Reviewer #2:

This manuscript investigates the role of HSP90/3 in Sonic hedgehog (SHH) mediated NASH development. The authors extend their previous studies on investigating SHH signaling mechanisms to identify downstream mediators. Here the investigators show that HSP90/3 is critical in SHH mediated fatty liver development and inflammation. The authors utilize in vivo and in vitro strategies to unravel the role of SHH-HSP90/3-miR-28-5p axis in NASH, however several concerns listed below, in methodologies and lack of clarity in results presented, dampen enthusiasm. Overall the data presented lack robust outcomes to justify hepatocyte specific SHH-HSP90/3 signaling in NASH.

We are very grateful to the reviewer for reviewing the manuscript carefully. We performed the experiments as suggested to address the reviewer's concerns.

1)Experiments in Figure 1 utilize cyclopamine, pharmacological inhibitor of SMO, to determine the effect of SHH signaling on murine NASH development. Using this inhibitor, investigators also evaluate alterations in HSP90/3 in Figure 2. Additional specific inhibition strategies of SHH signaling such as siRNA must be used for robust analysis, since identification of SHH-specific HSP90/3 changes is based on this inhibition.

We performed siRNA experiments as the reviewer suggested, the results were added in the revised manuscript. Similar as Smo inhibitor, knocking down Smo resulted in decreased Hsp90 β expression by promoting its ubiquitylation (revised manuscript, Fig. 2j-k, and 4d).

2)In Figure 2, in vivo and in vitro approaches show that HSP90 β protein is increased by SHH in HFFC-livers and primary hepatocytes as well as HEPA 1-6 cells and HEPG2 cells. However, it is not clear if this increase

in HSP90/3 occurs in hepatocytes only. Whether HSP90/3 is altered in liver macrophages and stellate cells must be evaluated. It is important to demonstrate cell-type specificity in HSP90/3 to justify subsequent generation of hepatocyte-specific HSP90/3 KO mice.

We checked the expression of Hsp90j3 in the isolated hepatocytes, Kupffer cells and hepatic stellate cells from mice liver. Then we checked the Hsp90 expression in the presence of Shh or compared between normal chow diet and HFFC diet treatment. Hsp90j3 were only upregulated in hepatocytes from HFFC-fed mice liver. Similarly, Hsp90j3 expression were only upregulated by Shh treatment in hepatocytes but not changed in Kupffer cells and hepatic stellate cells (revised manuscript, Fig. 2c and d). Taken together, these data suggested that Hsp90j3 overexpression was hepatocyte-specific in a NASH mouse model. Thus, we generated hepatocyte-specific Hsp90j3 KO mice to evaluate its role in NASH development.

3) Figure 2 I-N shows HSP90/3 and HSP90a levels in human liver samples. It is not clear if the liver samples are NAFLD or NASH patients? Also lack of representative IHC micrographs make it difficult to discern the levels shown in the graphs.

We thank the reviewer to point this out. We re-organized all the clinical data, and categorized into three different groups: healthy controls, NAFLD and NASH patients. When compared among these groups, the expression levels of HSP90j3 were positively correlated with disease progression, while the expression of HSP90a was not related to disease progression (revised manuscript, Fig. 2m-o).

4) HSP90/3 flox/flox mice were generated commercially. Data must be shown to confirm specificity of these mice. Further to genotype hepatocyte-specific HSP90/3 KO mice Fig S2B confirms that HSP90/3 is not expressed in liver tissues of KO mice. This is intriguing since

HSP90/3 is a constitutive form of HSP90 and expressed in several cell types in the liver including macrophages. To confirm hepatocyte specific KO, HSP90/3 must be decreased or lost in hepatocytes whereas other liver cells should exhibit HSP90/3 expression.

The western blot bands were developed with short exposure time. When exposed for a longer period, we could see Hsp90j3 bands in livers of *Hsp90/3^{3Hep}* mice (Response Figure 1 and revised manuscript, Figure S2b). We further verified the expression levels of Hsp90j3 protein in hepatocytes, Kupffer cells and hepatic stellate cells, isolated from *Hsp90/3^{3Hep}* mice. The results showed that only hepatocyte Hsp90j3 was eliminated (revised manuscript, Fig. S2c).

Response Figure 1 Western blot analysis of Hsp90j3 in the liver tissues of male or female *Hsp90j3^{fl/fl}* and *Hsp90/3^{3Hep}* mice.

5) Figure S5 utilizes 17-AAG in vivo as a HSP90 inhibitor to confirm the specificity of HSP90/3 in murine NASH. 17-AAG can inhibit HSP90a and HSP90/3 in various cell types. Additional strategies to target HSP90/3 using siRNA must be employed to determine SHH signaling in hepatocytes.

Figure S5 demonstrates that low-dose 17AAG improved the NASH phenotype, similar to that in *Hsp90/3^{3Hep}* mice. In this manuscript, we found that Hsp90j3 was mediated by SHH pathway. However, as reviewer mentioned, whether SHH signaling is affected by Hsp90j3 is unknown. Instead of using siRNA, we isolated primary hepatocytes from *Hsp90/3^{3Hep}* mice. It

turned out that there was no significance of *Gli1* gene expression in primary hepatocytes (Response Figure 2). Taken all our data together, we concluded that Hsp90 β was the downstream of SHH signaling pathway.

Response Figure 2 *Gli1* gene expression in primary hepatocytes of *Hsp90 $\beta^{fl/fl}$* and *Hsp90 $\beta^{\Delta Hep}$* mice.

6) Overexpression of SHH using ADV-SHH was demonstrated by detection of GFP+ signals in livers (Fir S6A). It is not clear which cells overexpress SHH and the magnitude of overexpression?

ADV virus is easy to infect hepatocytes and Kupffer cells, while it is hard to infect immunity cells (B cells, T cells, etc) and HSCs. As Liu Qiongmeng investigated, sh-control or Pu.1 adenovirus transduces both hepatocytes and Kupffer cells without affecting Pu.1 expression in B cells, T cells, or HSCs (Liu et al., 2020). Thus, in order to confirm the infection and expression of ADV in the liver, we evaluate their infection efficiency by flow cytometry. We found that in all the GFP+ cells, around 84.8% expressed albumin, a hepatocyte marker, while 7.26% GFP+ cells expressed F4/80, a macrophage marker. These results suggested that most of the cells that overexpressed Shh are hepatocytes.

Response Figure 3 Mice were administered ADV-Shh-GFP. Liver cells were analyzed by flow cytometry to evaluate their infection efficiency.

7) Much of the work on HSP90/3 deubiquitylation is performed in hepatocyte cell lines. These studies must be confirmed in primary hepatocytes and changes in HSP90/3 deubiquitylation must be demonstrated in NASH hepatocytes. EMSA analysis using BMAL1 antibodies (Fig 5D) is not convincing.

As the reviewer suggested, we performed the Hsp90j3 deubiquitylation experiments in primary hepatocytes. As expected, Usp31 knockdown reversed the reduction of Hsp90j3 ubiquitylation (revised manuscript, Fig. 4i and j). *In vitro* ubiquitination experiments showed that Hsp90j3 ubiquitylation decreased when incubated with the primary hepatocyte lysates treated with Shh or isolated from HFFC-diet mice (revised manuscript, Fig. 4e and S7f). The Hsp90j3 ubiquitylation in primary hepatocytes isolated from HFFC-diet mice also decreased (revised manuscript, Fig. 4c). EMSA analysis was re-performed in primary hepatocytes and replaced the previous data (revised manuscript, Fig. 5d).

8) SHH induced inflammation in NASH is attributed to HSP90 β facilitated extracellular communication by liver exosomes. The protocol used to prepare liver exosomes is not clear. This group has previously published preparation of exosomes from adipose tissue and the same method is extended to the liver. It is not clear if these purified liver exosomes capture the disease related phenotype.

A reference (Matejovič et al., 2021) to the protocol for extracting exosomes from the liver described in our method was not included. In this reference, three protocols of EV extraction from livers were compared. It turned out that the Protocol C, which was used in this manuscript, seemed to contain a higher amount of smaller size EVs (mean vesicle size: 117 \pm 40.6 nm) with an EV-like morphology. We also referred to this article, as Protocol C perfectly met our needs, and the method was consistent with that from adipose tissues we quoted.

Reference

Liu Q, et al. Inhibition of PU.1 ameliorates metabolic dysfunction and non-alcoholic steatohepatitis. *J Hepatol* 73, 361-370 (2020).
Matejovič A, Wakao S, Kitada M, Kushida Y, Dezawa M. Comparison of separation methods for tissue-derived extracellular vesicles in the liver, heart, and skeletal muscle. *FEBS Open Bio* 11, 482-493 (2021).

Reviewer #3 (Remarks to the Author):

Your elegant studies reveal a novel Hedgehog signaling mechanism that has potentially broad implications, including induction of a pro-inflammatory macrophage phenotype that contributes to the pathogenesis of nonalcoholic steatohepatitis (NASH) and related liver fibrosis.

Briefly, your work complements and extends earlier work by other groups showing that both hepatic expression of SHH ligand and SHH-initiated down-stream signaling that activates Smoothed (Smo)

are increased in people and mice with NASH. Importantly, your studies show that systemic administration of cyclopamine (Cp), a direct inhibitor of Smo, improves diet-induced steatosis, inflammation and fibrosis in mouse models. Although not cited in your manuscript, the inhibitory effects of direct Smo antagonism on murine liver inflammation and fibrosis had been reported previously and attributed to paracrine effects of hepatocyte-derived HH ligands on other types of HH-responsive liver cells (PMID: 21912653). However, the earlier study did not investigate the possible role of autocrine hepatocyte HH signaling in NASH pathogenesis.

Your work demonstrates that SHH-initiated activation of Smo in hepatocytes permits Bmal1 (a circadian clock-associated transcription factor) to disassociate from SuFu. Your result complements and extends data from another group which showed that hepatocyte Smo activity regulates the circadian clock (PMCID PMC6146234). You have significantly advanced understanding of the mechanisms involved by demonstrating that following Smo activation and release from SuFu, "free" Bmal1 localizes to nuclei where it promotes the transcription of a deubiquitinase (Usp 31) that stabilizes the heatshock chaperone, Hsp90b. Hsp90b, in turn, facilitates the release of hepatocyte-derived exosomes carrying miR 28-5p and this miR suppresses Rap1b in macrophages, resulting in activation of NF-KB signaling and increased macrophage production of several pro-inflammatory cytokines that are known to promote insulin resistance, hepatic steatosis, lipotoxicity and fibrogenic activation of hepatic stellate cells. Based on these results and evidence that SHH and mir28-5p are increased in humans with NASH, you suggest that miR28-5p might be both a novel biomarker and therapeutic target in NASH. The data are compelling and as you state, have actionable implications clinically. However, a few issues require clarification:

1 - Fig 2l is confusing as presented. More granular information is necessary to assess correlations between expression of SHH and Hsp90B and NASH severity. For example, how do levels of SHH/Hsp90b correlate with the severity of the NAFLD Activity Score (NAS) and fibrosis stage? The NAS is typically assessed to judging the severity of hepatic steatosis (graded 0-3), liver inflammation (graded 0-2) and hepatocyte ballooning (graded 0-2). Fibrosis staging (generally scored 0-4) is used to assess fibrosis severity. This information is particularly important for your studies because SHH is known to accumulate in ballooned hepatocytes and the severity of hepatocyte ballooning and liver fibrosis severity are significantly correlated.

We thank the reviewer to point this out. We re-organized all the clinical data, and categorized into three different groups: healthy controls, NAFLD and NASH patients. When compared among these groups, the expression levels of HSP90 β were positively correlated with disease progression, while the expression of HSP90 α was not related to disease progression (revised manuscript, Fig. 2m-o).

2 - You suggest that circulating miR28-5p might be a novel biomarker for NASH. You also show that serum levels of SHH increase in NASH and that SHH initiates the process that leads to increased miR28-5p. Is miR28-5p superior to SHH for "predicting" severity of NASH or fibrosis?

When extracting exosomes from clinical human serum samples, we also tested the concentration of SHH in the serum with an ELISA kit. Although the average concentration of SHH in the serum of NASH patients was higher than healthy controls, but there was no significant difference between the two groups (revised manuscript, Fig. 6i). In normal group, SHH concentration of 4 volunteers was far higher than others in the same group. As reported, SHH secretion elevated not only in NASH but also in pulmonary fibrosis (Kugler et al., 2015), chronic kidney disease (Zhou et al., 2016), and alcohol-associated

liver injury (Kumar et al., 2023). Therefore, we think that miR-28-5p in exosomes from serum serves as a better marker to predict the severity of NASH patients.

3 - Some of the approaches that you used to manipulate HH signaling (e.g., systemic administration of Cp or Adenoviral vector delivery of SHH ligand) are not hepatocyte-specific and so, would impact pathway activity in multiple cell types both in liver and other tissues. While your in vitro experiments clearly demonstrate that SHH can induce Hsp90b in hepatocytes and your in vivo experiments prove that hepatocyte derived Hsp90b critically mediates liver inflammation and fibrosis, can you assure that SHH is the main mediator of Hsp90b accumulation in hepatocytes in your mouse NASH models (or human NASH liver samples)? What was the rationale for using ADV-SHH in mice with diet-induced NASH given that your data and work of others shows that endogenous SHH expression is increased in NASH?

In vitro, Shh ligand promotes the expression of Hsp9013, in primary hepatocytes, Hepa 1-6 and AML12 cells (revised manuscript, Fig. 2d-i). *In vivo*, Shh overexpression in liver promoted the Hsp9013 protein levels (revised manuscript, Fig. 3b). Before we focused on Shh, we checked a series of NASH risk factors, such as PA and OA, cytokines IL-113, cytokines TNF-a, LPS, hydrogen peroxide, glucose and insulin, in regulating Hsp9013 protein levels in Hepa 1-6 cells, none of the treatments promoted Hsp9013 protein levels (revised manuscript, Fig. S7a and b). Thus, we believed that Shh played an very important role in mediating Hsp9013 expression.

Normally, HFFC diet feeding for 8 weeks did not induce NASH-related inflammation and fibrosis phenotypes. Therefore, we try to figure out whether Shh could accelerate the process of NASH. As we expected, in *Hsp90β^{fl/fl}* mice, ADV-Shh accelerated hepatic damage (revised manuscript, Fig. S6f-g), inflammation (revised manuscript, Fig. 3d-f), and fibrosis (revised manuscript,

Fig. 3g-h) in HFFC-fed mice for only 8 weeks. These phenotypes were prevented by hepatic ablation of Hsp90 β (revised manuscript, Fig. S6f-g, 3d-f and 3g-h).

4 - Previous publications have shown that liver cell-derived exosomes carry SHH and Indian Hedgehog (PMCID PMC3724240). Studies have also reported that hepatocyte-derived IHH is important for activating macrophages and promoting hepatic stellate cell activation in mouse models of NASH (PMCID PMC5226184). Do the exosomes that carry miR25p also carry either of the HH ligands? If so, is the HH ligand cargo required to miR28-5p to promote the proinflammatory macrophage phenotype?

Shh exists in exosomes derived from livers of normal or NASH mice. In addition, the Shh ligand is also known to stimulate hepatic stellate cells and fibroblasts via a paracrine mechanism, thereby promoting profibrogenic response in mouse model of NASH (Kwon et al., 2016; Rangwala et al., 2011; Jung et al., 2010; Choi et al., 2009; Hirsova et al., 2013; Omenetti et al., 2008).

Similar to that (PMCID PMC5226184), Ihh, secreted by hepatocytes, activates fibrogenic genes in hepatic stellate cells (Wang et al., 2016). However, this article only proves Ihh activated HSC, but did not mention whether Ihh activated macrophages.

To study the effects of Shh on macrophages, BMDM cells were treated with Shh, no proinflammatory phenotypes were observed (revised manuscript, Fig. 6a). In contrast, when BMDM cells were treated with miR-28-5p mimics alone, the gene expression of pro-inflammatory cytokines was increased (revised manuscript, Fig. 6e). Taken together, we believe that miR-28-5p, but not Shh, promoted the proinflammatory macrophage phenotypes.

4 - Can you rectify your findings which suggest that stimulating Smo activity in hepatocytes promotes hepatic lipid accumulation with recent

work from at least two other groups demonstrating that targeted inhibition of hepatocyte Smo promotes hepatocyte lipid accumulation (PMCID PMC4869931, PMID PMC8559255)?

Of the two recent works reviewer mentioned, one of the articles (PMID PMC8559255) could not be found, probably because of the wrong PMC number. In another work (Matz-Soja et al., 2016), hepatocyte-specific Smo deletion mice of 8 weeks old were fed with normal diet. In adult healthy liver, SHH signaling is inactive (Hirose et al., 2009; Machado and Diehl, 2018), thus, the effect of Smo KO would not significantly affect Shh signaling pathway further. Shh is reactivated when the liver is injured, or HFFC-diet feeding as described in our work. This article mentioned that “The potential of impaired Hh signaling to trigger steatosis independent of nutritional changes suggests that malfunctions in this pathway may pave the way for the development of NAFLD long before other cues may lead to further aggravation.” This statement might not reflect the progression of NASH, since Shh expression is higher in livers from both NASH patients and NASH animal models.

In our manuscript, we found that Shh increased Hsp90j3 protein levels in hepatocytes, while studies have showed that Hsp90j3 promotes *de novo* lipid synthesis in hepatocytes by increasing the expression of SREBP (Zheng et al., 2019; Kuan et al., 2017). Therefore, we think lipid accumulation reduction in our report is mainly due to the reduced SREBPs and *de novo* lipogenesis in the liver.

5 - Others have reported that Smo activity inhibits adipogenesis but you found that systemic exposure to Cp (a Smo antagonist) reduced adiposity in mice with diet-induced obesity and NASH. How might this paradox be explained?

As the reviewer did not provide the PMCID of the research, we find an article closed to this (Suh et al., 2006). In vitro, SHH treatment in 3T3-L1 cells prevented adipogenesis. At the same time, the authors found that in

HFD-induced obese mice, the gene expression of *Smo*, *Gli1*, *Gli2*, and *Gli3* were significantly decreased, indicating that SHH pathway was inhibited. In the absence of Shh, Smo activity is repressed by PTCH1 (Teperino et al., 2014), therefore under these conditions, administration of Cp (a Smo antagonist) might not cause significant effects on SHH pathway.

We further checked the distribution of Cp *in vivo*. Cp was injected intraperitoneally, after 1 hour, Cp concentration was detected with HPLC-mass spectrometry. Using semi-quantitative detection of peak areas, it was found that the concentration of Cp in adipose tissues was only 24.4% of that in the liver (Response Figure 4). These data suggested that Cp mainly function in the liver.

In this manuscript, we also found that Cp prevented lipid accumulation by decreased *de novo* lipid synthesis in liver and we believed that the weight loss of mice after Cp treatment was mainly because of that.

Response Figure 4 Mice were injected intraperitoneally with Cp and its concentration was detected with HPLC-mass spectrometry.

References:

- Kugler MC, Joyner AL, Loomis CA, Munger JS. Sonic hedgehog signaling in the lung. From development to disease. *Am J Respir Cell Mol Biol* 52, (2015). Zhou D, Tan RJ, Liu Y. Sonic hedgehog signaling in kidney fibrosis: a master communicator. *Sci China Life Sci* 59, 920-929 (2016).
- Kumar V, et al. Anti-miR-96 and Hh pathway inhibitor MDB5 synergistically ameliorate alcohol-associated liver injury in mice. *Biomaterials* 295, 122049 (2023).
- Kwon H, et al. Inhibition of hedgehog signaling ameliorates hepatic inflammation in mice with nonalcoholic fatty liver disease. *Hepatology* 63, 1155-1169 (2016).
- Rangwala F, et al. Increased production of sonic hedgehog by ballooned hepatocytes. *J Pathol* 224, 401-410 (2011).
- Jung Y, et al. Signals from dying hepatocytes trigger growth of liver progenitors. *Gut* 59, 655-665 (2010).
- Choi SS, et al. Hedgehog pathway activation and epithelial-to-mesenchymal transitions during myofibroblastic transformation of rat hepatic cells in culture and cirrhosis. *Am J Physiol Gastrointest Liver Physiol* 297, G1093-G1106 (2009).
- Hirsova P, Ibrahim SH, Bronk SF, Yagita H, Gores GJ. Vismodegib suppresses TRAIL-mediated liver injury in a mouse model of nonalcoholic steatohepatitis. *PLoS One* 8, e70599 (2013).
- Omenetti A, et al. Hedgehog signaling regulates epithelial-mesenchymal transition during biliary fibrosis in rodents and humans. *J Clin Invest* 118, 3331-3342 (2008).
- Wang X, et al. Hepatocyte TAZ/WWTR1 Promotes Inflammation and Fibrosis in Nonalcoholic Steatohepatitis. *Cell Metab* 24, 848-862 (2016).
- Matz-Soja M, et al. Hedgehog signaling is a potent regulator of liver lipid metabolism and reveals a GLI-code associated with steatosis. *Elife* 5, (2016).
- Hirose Y, Itoh T, Miyajima A. Hedgehog signal activation coordinates proliferation and differentiation of fetal liver progenitor cells. *Exp Cell Res* 315, 2648-2657 (2009).
- Machado MV, Diehl AM. Hedgehog signalling in liver pathophysiology. *J Hepatol* 68, 550-562 (2018).
- Zheng Z-G, et al. Inhibition of HSP90 β Improves Lipid Disorders by Promoting Mature SREBPs Degradation via the Ubiquitin-proteasome System. *Theranostics* 9, 5769-5783 (2019).
- Kuan Y-C, et al. Heat Shock Protein 90 Modulates Lipid Homeostasis by Regulating the Stability and Function of Sterol Regulatory Element-binding Protein (SREBP) and SREBP Cleavage-activating Protein. *J Biol Chem* 292, 3016-3028 (2017).
- Suh JM, Gao X, McKay J, McKay R, Salo Z, Graff JM. Hedgehog signaling plays a conserved role in inhibiting fat formation. *Cell Metab* 3, 25-34 (2006).

Teperino R, Aberger F, Esterbauer H, Riobo N, Pospisilik JA. Canonical and non-canonical Hedgehog signalling and the control of metabolism. *Semin Cell Dev Biol* 33, 81-92 (2014).

REVIEWER COMMENTS

Reviewer #1 (Remarks to the Author):

In this revised manuscript, the authors have done plenty of experiments including detection of miR-28-5p in human samples . They have successfully addressed my concerns and I have no further questions.

Reviewer #3 (Remarks to the Author):

Your work identifies a novel mechanism whereby Sonic hedgehog ligand (SHH) activates Smoothed and Gli1-dependent canonical hedgehog signaling in liver cells to promote the pathogenesis of NASH. Reviewers of the initial version of the manuscript had some questions and concerns that you have largely addressed via additional experiments. This information has been incorporated into the revised manuscript and provides additional support for your model. I have not further concerns.

Reviewer #4 (Remarks to the Author):

In this study, Zhang et al. suggest that the activation of the Sonic hedgehog pathway promotes Hsp90 β stability and causes NASH.

Many results, however, argue against the authors' hypothesis.

For instance, the authors state "We further checked Hsp90 β expression isolated primary hepatocytes (HPs), Kupffer cells (KCs) and hepatic stellate cells (HSCs) from mice fed with chow or HFFC diet for 16 weeks. It turned out that Hsp90 β only increased in primary hepatocytes (Fig. 2c). Likewise, Shh only promoted Hsp90 β protein levels in HPs".

Indeed, by definition NASH is a histological diagnosis made when steatosis \geq 5%, hepatocyte ballooning and inflammation are simultaneously present. Moreover, liver fibrosis staging 1 to 3 is a concomitant feature. Instead, Hsp90 β was found to increase only in primary hepatocytes suggesting no role of this heat shock protein in inflammation and in fibrosis.

The authors observe “We found that HSP90 β was positively correlated with disease progression, while the expression of HSP90 α was not related to disease progression (Fig. 2m–o)”. Indeed, in Figure 2 panel n the HSP90 β levels were similar in NAFL and NASH, therefore I could not appreciate a “positive correlation with disease progression”.

The models used in this study do not replicate what happens in the liver during NASH development. Instead of doing hepatocyte Hsp90 β ablation to observe effects on steatohepatitis, a good study design would have included organoids where tissue architecture is maintained and interactions/crosstalk between different cell types is present.

What does “Liver tissues from Hsp90 β fl/fl or Hsp90 β Δ Hep 648 mice were obtained” mean? Reading carefully the manuscript, I realized that the authors worked only on isolated cells where the gene codifying for HSP90 β was suppressed or overexpressed. Thus, they did not use conditional KO or transgenic mice.

“Gene knockdown and overexpression

For transient transfection, cells were transfected with indicated miRNA or siRNA constructs (Genepharma, Shanghai, China) using RNAiMAX (Invitrogen, California, USA), or indicated plasmids using the Lipofectamine 3000 (Invitrogen, California, USA) reagent following the manufacturer’s instructions.”

Overall, this study does not show causality but just correlations.

In addition, it is clear that the manuscript was prepared for another journal where methods precede the results. In fact, the results section contains a series of abbreviations that are reported only in the methods. One for all, “high fat, high fructose, and high cholesterol diet (HFFC)”.

Finally, please do not address people as “obese”, like in lines 76 and 77 “patients and obese mice” and “overweight and obese children”, but rather say children with overweight or obesity, this to avoid obesity stigma.

*We acknowledge all the reviewers' efforts to improve our manuscript. As the review 1 and 3 are satisfactory with our response and revised manuscript, while the reviewer 2 did not respond during the first revision round, here, we have attempted to address the reviewer 4' concerns point-to point. The reviewer 4' comments are provided in **bold italics** below with our response following.*

Reviewer #4 (Remarks to the Author):

In this study, Zhang et al. suggest that the activation of the Sonic hedgehog pathway promotes Hsp90/3 stability and causes NASH.

Many results, however, argue against the authors' hypothesis.

We appreciate the reviewer's deeply insightful comments. We explained all the questions reviewer concerned and demonstrate the existence of intercellular communication through organoid experiments as reviewer suggested.

1. For instance, the authors state "We further checked Hsp90/3 expression isolated primary hepatocytes (HPs), Kupffer cells (KCs) and hepatic stellate cells (HSCs) from mice fed with chow or HFFC diet for 16 weeks. It turned out that Hsp90/3 only increased in primary hepatocytes (Fig. 2c). Likewise, Shh only promoted Hsp90/3 protein levels in HPs".

Indeed, by definition NASH is a histological diagnosis made when steatosis $\geq 5\%$, hepatocyte ballooning and inflammation are simultaneously present. Moreover, liver fibrosis staging 1 to 3 is a concomitant feature. Instead, Hsp90/3 was found to increase only in primary hepatocytes suggesting no role of this heat shock protein in inflammation and in fibrosis.

In this manuscript, we discovered that HSP90 β only elevated in hepatocytes and played an indirect role in promoting inflammation and fibrosis. In murine and human liver cell lines and primary hepatocytes,

Hsp90P was increased after Shh treatment (Fig. 2g-i). The expression levels of HSP90P were also increased in NASH patients, compared with healthy controls (Fig. 2n-o). *In vivo*, hepatocyte-specific Hsp90P ablation reversed inflammation and fibrosis in the liver induced by HFFC (Supplementary Fig. 4a-e). To understand the mechanism how hepatic HSP90P affected the inflammation in macrophages, we analyzed the cell-cell communications in the animal models. It turned out that the increased HSP90P protein in hepatocytes promoted the secretion of exosomes enriched with miR-28-5p, which stimulated NF- κ B transcriptional activity in macrophages and increased the expression of inflammatory factors (Fig. 6c-e). *In vivo*, targeted delivery of miR-28-5p antagomir to hepatic macrophages mitigated the HFFC-induced hepatic damage and inflammation, which suppressed the development of NASH (Fig. 8b-k). Taken together, as the reviewer suggested, hepatic Hsp90P has no direct effects on inflammation and fibrosis, but rather through exosomes enriched with miR-28-5p.

2. The authors observe “We found that HSP90/3 was positively correlated with disease progression, while the expression of HSP90a was not related to disease progression (Fig. 2m–o)”. Indeed, in Figure 2 panel n the HSP90/3 levels were similar in NAFL and NASH, therefore I could not appreciate a “positive correlation with disease progression”.

We thank reviewer for pointing this out. The average IHC intensity of HSP90P was 14.10 in normal volunteers, 32.86 in NAFL patients and 37.90 in NASH patients, indicating that hepatic HSP90P levels in NASH were high than that in NAFL. The p value between NAFL and NASH patients group was 0.0828, very close to statistic significance. In this manuscript, we discovered that HSP90P was important for the development of NAFLD. The upregulation of Hsp90P in hepatocytes promoted the secretion of exosomes enriched with miR-28-5p, which

promoted NF- κ B transcriptional activity in macrophages and stimulated the expression of inflammatory factors. In addition, mice hepatic ablation of Hsp90 β prevented HFFC-induced hepatic steatosis, inflammation, and fibrosis. Thus, Hsp90 β had indirect effects on inflammation and fibrosis, thereby promoting the development of NAFLD. As the reviewer pointed out, the sentence “positive correlation with disease progression” is not very accurate. We have changed it to “We found that HSP90 β was important for the development of NAFLD, while the expression of HSP90 α was not related to disease progression (Fig. 2m–o)”.

3. The models used in this study do not replicate what happens in the liver during NASH development. Instead of doing hepatocyte Hsp90 β ablation to observe effects on steatohepatitis, a good study design would have included organoids where tissue architecture is maintained and interactions/crosstalk between different cell types is present.

Extracellular vesicles (EVs) act as a signaling mediator, resulting in lipid accumulation, macrophage and hepatic stellate cell activation, further promoting inflammation and liver fibrosis progression during the development of NAFL/NASH (Xu X, Poulsen KL, Wu L, Liu S, Miyata T, Song Q, Wei Q, Zhao C, Lin C, Yang J. Targeted therapeutics and novel signaling pathways in non-alcohol-associated fatty liver/steatohepatitis (NAFL/NASH). *Signal Transduct Target Ther.* 2022 Aug 13;7(1):287.). In this manuscript, we discovered that exosomes enriched with miR-28-5p secreted by hepatocytes promoted the expression of inflammatory factors in macrophages. The crosstalk between hepatocytes and macrophages was proved by extracting liver exosomes from Hsp90 β fl/fl or Hsp90 β Δ Hep mice with chow or HFFC diet, and then detecting the inflammatory factors in BMDM cells after treated with hepatocytes-derived exosomes (Fig. 6c). We also performed liver organoids experiments as the reviewer suggested. It turned out that the

results were in consistent with cell co-culture experiments. Shh promoted inflammatory gene expression in organoid derived from Hsp90 $\beta^{fl/fl}$ mice. Increased Il-1 β , Tnf- α and Il-6 expression could be blunted either in hepatic organoid derived from liver-specific Hsp90 knockout mice (Hsp90 $\beta^{\Delta Hep}$), or in hepatic organoid derived from Hsp90 $\beta^{fl/fl}$ mice treated with miR-28-5p antagomir (Supplementary Figure 11).

4. What does “Liver tissues from Hsp90/3 fl/fl or Hsp90/3 Δ Hep 648 mice were obtained” mean? Reading carefully the manuscript, I realized that the authors worked only on isolated cells where the gene codifying for HSP90/3 was suppressed or overexpressed. Thus, they did not use conditional KO or transgenic mice.

In this manuscript, we generated tissue-specific KO mice (Hsp90 $\beta^{\Delta Hep}$) by crossing Hsp90 $\beta^{fl/fl}$ with Alb-Cre mice (Supplementary Fig. 2). Hsp90 β levels in various cell types from livers were examined after genotyping (Supplementary Fig. 2b and c). The genotype of 18 mice for the experiments was confirmed (Supplementary Fig. 2d). Hsp90 $\beta^{fl/fl}$ and Hsp90 $\beta^{\Delta Hep}$ mice were used to discover the role of Hsp90 β in NASH development (Fig. 3a-h and Supplementary Fig. 3a-k, 4a-e and 6a-g). The paragraph that the reviewer mentioned is to describe how we isolate exosomes from mice hepatocytes with different genetic background. The aim of this study is to investigate the cell-cell communications via secreted exosomes. Hepatic Hsp90 β per se did not affect inflammation response, however, Hsp90 β facilitate the secretion of exosomes from the liver that were enriched with miR-28-5p, miR-28-5p then activated the inflammatory response in macrophages (Fig. 6b-f).

5. “Gene knockdown and overexpression

For transient transfection, cells were transfected with indicated miRNA

or siRNA constructs (Genepharma, Shanghai, China) using RNAiMAX (Invitrogen, California, USA), or indicated plasmids using the Lipofectamine 3000 (Invitrogen, California, USA) reagent following the manufacturer's instructions." Overall, this study does not show causality but just correlations.

This part of the method has been corrected to "To demonstrate the impacts of shh pathway on HSP90 β protein levels, primary hepatocytes, Hepa 1-6 or HepG2 cells were transfected with indicated siRNA (Genepharma, Shanghai, China) using RNAiMAX (Invitrogen, California, USA) following the manufacturer's instructions. To perform co-immunoprecipitation or in vitro deubiquitination assay, indicated plasmids were transfected with Lipofectamine 3000 in primary hepatocytes, Hepa 1-6 or 293T cells following the manufacturer's instructions. To study the role of miR-28-5p in regulating inflammation in macrophages, BMDM cells were transfected with miR-28-5p mimic or miR-28-5p antagomir using RNAiMAX (Invitrogen, California, USA) following the manufacturer's instructions."

6. In addition, it is clear that the manuscript was prepared for another journal where methods precede the results. In fact, the results section contains a series of abbreviations that are reported only in the methods. One for all, "high fat, high fructose, and high cholesterol diet (HFFC)". Finally, please do not address people as "obese", like in lines 76 and 77 "patients and obese mice" and "overweight and obese children", but rather say children with overweight or obesity, this to avoid obesity stigma.

We thank reviewer for pointing these out. We have corrected these issues as the reviewer suggested.

REVIEWER COMMENTS

Reviewer #4 (Remarks to the Author):

The concerns expressed in my previous review are still present, this because the responses of the authors utterly were unconvincing.

The authors generated hepatic Hsp90 β knockout mice (Hsp90 β Δ Hep), this resulted however in the absence of Hsp90 β expression only in hepatocytes but not in Kupffer or in HSC, as shown in Figure 2 in the supplementary material.

The authors state “miR-28-5p directly targeted and decreased Rap1b levels, which in turn promoted NF- κ B transcriptional activity in macrophages and stimulated the expression of inflammatory factors.” There is no evidence that miR-28-5p promote differentiation of HSC into myofibroblasts nor that it induces hepatocyte ballooning. As I previously commented, NASH results from the simultaneous presence of liver steatosis >5%, hepatocyte ballooning and inflammation. These features are not demonstrated in this study, hence the conclusions of the authors are erroneous.

As a minor comment, the authors affirm “The p value between NAFL and NASH patients group was 0.0828, very close to statistical significance.” Indeed, this p value is clearly not significant.

*We acknowledge the reviewers #4' additional efforts to improve our manuscript, here, we have attempted to address the reviewer 4' concerns point-to point. The reviewer 4' comments are provided in **bold italics** below with our response following.*

REVIEWER COMMENTS

Reviewer #4 (Remarks to the Author):

The concerns expressed in my previous review are still present, this because the responses of the authors utterly were unconvincing.

The authors generated hepatic Hsp90/3 knockout mice (Hsp90/3 Δ Hep), this resulted however in the absence of Hsp90/3 expression only in hepatocytes but not in Kupffer or in HSC, as shown in Figure 2 in the supplementary material.

We are very grateful for the thorough review, which helped us a lot in the logic and language accuracy of this manuscript. “hepatic Hsp90 β knockout mice (Hsp90 $\beta^{\Delta H_e p}$)” used in this manuscript is not accuracy. We generated hepatocellular Hsp90 β knockout mice by crossing hepatocyte specific Alb-cre and Hsp90 β flox mice. Hsp90 β is deficient in hepatocytes, without changing the Hsp90 β levels in Kupffer cells or in HSC. Therefore, we correct all the inaccurate expression “hepatic Hsp90 β knockout mice” with “hepatocellular Hsp90 β knockout mice”.

The authors state “miR-28-5p directly targeted and decreased Rap1b levels, which in turn promoted NF- κ B transcriptional activity in macrophages and stimulated the expression of inflammatory factors.” There is no evidence that miR-28-5p promote differentiation of HSC into myofibroblasts nor that it induces hepatocyte ballooning. As I previously commented, NASH results from the simultaneous presence of liver

steatosis >5%, hepatocyte ballooning and inflammation. These features are not demonstrated in this study, hence the conclusions of the authors are erroneous.

In this manuscript, it was discovered that miR-28-5p in hepatic EVs was increased in NASH mice and downregulated in Hsp90 $\beta^{\Delta Hep}$ mice with HFFC diet, as was the hepatic fibrosis and hepatocyte ballooning (Response Figure 1: Revised manuscript, Figure 6d, f and S4a, c, e). As pointed out by reviewer, hepatocyte ballooning is a critical feature in NASH. Thus, we marked ballooning hepatocytes with black arrows on all H&E stained liver sections (Response Figure 2: Revised manuscript, Figure 1d, 3c, 8b and S4a, 5h). As suggested by reviewer, we also had HSC administrated with miR-28-5p, finding that miR-28-5p promoted the fibrogenic gene expression of *Col1 α 1*, *α -Sma*, *Tgfbr1* and *Timp1* (Response Figure 3: Revised manuscript, Figure S11e). The same conclusion was reached when organoids were treated with miR-28-5p (Response Figure 3: Revised manuscript, Figure S11d).

F6d

F6f

FS4a

FS4c

FS4e

Response Figure 1: F6d Exosomes were collected from livers of *Hsp90β^{fl/fl}* and *Hsp90β^{ΔHep}*

mice fed with chow or HFFC diet, the expression of exosomes containing miRNA were analyzed by sequencing, the heatmap of miRNA expression was shown. C represented

Hsp90β^{fl/fl} mice on a normal chow diet. M represented *Hsp90β^{fl/fl}* mice on a HFFC diet. KO represented *Hsp90β^{ΔHep}* mice on a HFFC diet. **F6f** Exosomes were collected from primary hepatocytes of *Hsp90β^{fl/fl}* and *Hsp90β^{ΔHep}* mice after the treatment with 400 pg/μL Shh for 48 hours, and relative expression of miR-28-5p was analyzed by qRT-PCR. **FS4a** Representative H&E stained FFPE liver sections and ORO stained frozen liver sections. Each arrow indicated hepatocyte ballooning. **FS4c** Representative Masson and Sirius red stained liver sections **FS4e** Expression of fibrogenic genes in livers of *Hsp90β^{fl/fl}* and *Hsp90β^{ΔHep}* mice on a normal chow or HFFC diet .

Response Figure 2: F1d Formalin fixed paraffin embedded (FFPE) liver sections were stained with hematoxylin eosin (H&E) and frozen liver sections were stained with oil red O (ORO), respectively. Each arrow indicated hepatocyte ballooning. **F3c** Representative H&E-stained FFPE liver sections and ORO-stained frozen liver sections. Each arrow indicated hepatocyte ballooning. **F8b** Representative H&E stained FFPE liver sections and ORO stained frozen liver sections. Each arrow indicated hepatocyte ballooning. **FS4a** Representative H&E stained FFPE liver sections and ORO stained frozen liver sections. Each arrow indicated hepatocyte ballooning. **FS5h** Representative H&E stained FFPE liver sections and ORO stained frozen liver sections. Each arrow indicated hepatocyte ballooning.

Response Figure 3: F11d Fibrogenic genes expression in hepatic organoids when administrated with miR-NC mimic and miR-28-5p mimic. **F11e** Fibrogenic genes expression in HSCs when administrated with miR-NC mimic and miR-28-5p mimic.

As a minor comment, the authors affirm “The p value between NAFL and NASH patients group was 0.0828, very close to statistical significance.” Indeed, this p value is clearly not significant.

The p value between NAFL and NASH patient groups of HSP90 β was 0.0828, not significant. This may be because of the small number of clinical samples. However, the important effects of HSP90 β were proved in this manuscript. Hsp90 β promoted hepatic inflammation through miR-28-5p, which promoted NF- κ B transcriptional activity in macrophages and stimulated the expression of inflammatory factors.

REVIEWERS' COMMENTS

Reviewer #4 (Remarks to the Author):

Only very minor comments. Please add among the limitations of this study that the sample size was inadequate to demonstrate a significant difference between Hsp90 β in NAFL and NASH patients ($p=0.0828$).

*We acknowledge the reviewers #4' additional efforts to improve our manuscript, here, we have attempted to address the reviewer 4' concerns point-to point. The reviewer 4' comments are provided in **bold italics** below with our response following.*

REVIEWER COMMENTS

Reviewer #4 (Remarks to the Author):

Only very minor comments. Please add among the limitations of this study that the sample size was inadequate to demonstrate a significant difference between Hsp90 β in NAFL and NASH patients (p=0.0828).

We are very grateful for reviewer's comments, which helps us improve this manuscript a lot. We have added the limitations of this study "However, the limitation of this study was that the sample size was inadequate to demonstrate a significant difference between Hsp90 β in NAFL and NASH patients (p=0.0828)." in discussion section as reviewer advised.